

# MOPREDAScentury: a long-term monthly precipitation grid for the Spanish mainland

Santiago Beguería[1], Dhais Peña-Angulo[2], Víctor Trullenque-Blanco[2], Carlos González-Hidalgo[2,3]

[1]Estación Experimental Aula Dei, Consejo Superior de Investigaciones Científicas (EEAD-CSIC), 50059 - Zaragoza, Spain.
[2]Departamento de Geografía y Ordenación del Territorio, Universidad de 50009 - Zaragoza, Spain.
[3]IUCA, Universidad de Zaragoza, 50009 - Zaragoza, Spain

*Correspondence to*: Santiago Beguería (santiago.begueria@csic.es)

**Abstract.** This article describes the development of a monthly precipitation dataset for the Spanish mainland (western Mediterranean basin), covering the period between December 1915 and December 2020. The dataset combines ground
observational data from the National Climate Data Bank (NCDB) of the Spanish national climate and weather service (AEMET) and new data rescued from meteorological yearbooks published prior to 1951 that was never incorporated into the NCDB. The yearbooks data represented a significant improvement of the dataset, as it almost doubled the number of weather stations available during the first decades of the 20[th] century, the period when the dataset was more scarce. The final dataset contains records from 11,312 stations, although the number of stations with data in a given month varies largely between 674
in 1939 and a maximum of 5,234 in 1975. Spatial interpolation was used on the resulting dataset to create monthly precipitation grids. The process involved a two-stage process: estimation of the probability of zero-precipitation (dry month), and estimation of precipitation magnitude. Interpolation was carried out using universal kriging, using anomalies (ratios with respect to the 1961-2000 monthly climatology) as dependent variable and several geographic variates as independent variables. Cross-validation results showed that the resulting grids are spatially and temporally unbiased, although the mean
error and the variance deflation effect are highest during the first decades of the 20[th] century, when the observational dataset was more scarce. The dataset is available at https://doi.org/10.20350/digitalCSIC/15136 under an open license, and can be cited as Beguería et al. (2023).



# 1 Introduction

Sea and land weather station records are crucial information sources to study the evolution of climate over the last century
and beyond, and are the result of the sustained effort of many volunteers and climate and weather agencies around the world
(see Strangeways, 2007). A large number of projects have focused on collecting and curating data from different sources in
order to improve the spatial and temporal coverage of the datasets, and even rescue old data that had not been digitised and
remains unknown to the broad public. These efforts are particularly needed in regions where water is a scarce and limiting
resource combined with a high demand, such as Mediterranean climate regions of the world. Especially in those areas,
however, research does not provide unanimous results, as for example trends analyses show differences according to the
period selected, data set or study area (Hoerling et al., 2012; Mariotti et al., 2015; Zittis, 2015; Deitch et al., 2017; Caloiero
et al., 2018; Peña-Angulo et al., 2020; between many others).

In parallel with these efforts, many research groups have focused on developing spatial and temporal complete grids that
override the fragmentary character of observational (station-based) data sets. The development of gridded climatic datasets
from point observations has experienced a fast development in the first decades of the $21^{st}$ century, aided by the tremendous
improvement of computing capabilities and the implementation of complex interpolation methods in standard statistical
packages and programming languages (New et al., 2002; Hijmans et al., 2005; Harris et al., 2014; Schamm et al., 2014;
Harris et al., 2020). Gridded data sets offer numerous advantages over point-based observational data that make them best
suited to climate and environmental studies. While observational datasets are limited to the locations of climatic stations and
the time series are often fragmentary in time, gridded datasets offer a continuous spatial and temporal coverage. Having a
continuous coverage is most relevant for computing regional or even global averages, which are crucial in climate change
studies. Gridded data are also ubiquitous in studies that involve the use of models, as they usually require continuous
climatic forcing data.

Users of gridded data, however, must not forget that grids are in fact models and not directly observed data, and as such they
are not devoid of issues. Interpolation methods are not perfect, and they have inherent problems such as the deflation of
(spatial and temporal) variance, as we discussed in Beguería et al. (2006). Also, since the spatial and temporal coverage of
observational datasets is often not homogeneous (some areas and time periods are over-represented while others may lack
any data), there are potential sources of bias. Despite this, gridded datasets are currently used in the vast majority of studies
that make use of climate data.

In a previous work we described the development of a gridded data set of monthly precipitation for Spain, MOPREDAS,
spanning 1946-2005 (González-Hidalgo et al., 2011). Other gridded precipitation data sets have been later developed for
Spain with a daily temporal resolution, such as Spain02 for the period 1950-2003 (Herrera et al., 2012) and SPREAD for
1950-2012 (Serrano-Notivoli et al., 2017). Currently, no data set exists spanning back to the first decades of the $20^{th}$ century.
This is due to the drastic decrease in the number of available observations prior to 1950. The objective of this article is to
describe the development of the MOPREDAScentury data set, a gridded data set of monthly precipitation over mainland





Spain covering the period 1916-2020, aimed at becoming the best spatio-temporal data set currently available to assess changes in the spatial and temporal patterns of precipitation over Spain. The process includes the rescue of old records not included in the digital database of the Spanish national weather service, which allow increasing the observational sample and are critical for developing a gridded dataset. The text describes this data rescue process and the spatial interpolation, presents

the main results of a cross-validation assessment, and discusses several issues related to the development of the dataset.

## 2 Data and methods

The development of the MOPREDAScentury dataset encompassed two distinctive steps: i) improving the observational dataset available in digital format, especially for the first half of the 20th century; and ii) using spatial interpolation techniques to create the gridded dataset. This section describes both steps, as well as the procedure used for evaluating the

resulting data set.

### 2.1 Data rescue (yearbooks)

The MOPREDAScentury data set combines land-based weather station data digitised and stored in the National Climate Data Bank (*Banco Nacional de Datos del Clima* in Spanish, BNDC hereinafter), and newly digitised records from

meteorological yearbooks (YB) that were published by different government offices until 1950 such as *Ministerio de Fomento*, *Servicio Meteorológico* (a part then of the *Instituto Geográfico y Catastral*) and *Ministerio del Aire*. The data rescue process from the yearbooks was carried out in two main steps: (a) digitisation, and (b) matching with the data series in the BNDC. Digitisation was carried out by using special flatbed scanners and manual reading and input. Matching the digitised data with the data series in the BNDC proved to be a laborious task, as the identification of the weather stations in

the YB were not consistent across the books and did not always coincide with the BNDC. Similar difficulties were found when rescuing temperature data to develop the MOTEDAScentury dataset (Gonzalez-Hidalgo et al., 2015 and 2020; a detailed description of the matching process can be found in these references). The rescued yearbooks data had a fair level of overlapping with the BNDC, but they allowed to fill in gaps and extend many time series back into the first decades of the 20th Century. There were also a number of data series that were completely new.

The augmented data set resulting from the combination of the BNDC and the YB rescued data was subjected to a manual quality control. Thus, the observations were automatically flagged as suspicious in the following cases:

- Sequences of twelve identical monthly values occurring in different years in the same station, or in the same year in different stations.
- Sequences of seven or more consecutive months with zero precipitation in the same station.

- Individual months with precipitation equal or greater than 1000 mm.




**The flagged data (suspicious values) were first checked in their original sources (books) to discard digitisation errors, and then they were compared against three or four neighbouring stations to decide whether to maintain or discard them. An example of data rejection is provided in**

Table 1.


## 2.2 Spatial interpolation (two-step method)

We use geostatistical techniques for the interpolation of monthly precipitation. Geostatistics is now a well-known field, and it has been presented in a wide range of introductory texts (Goovaerts, 1997), so we should provide only a brief summary here. The key element in geostatistics is the variogram (or, more commonly, the semivariogram), which is a function that

relates the semi-variance $\gamma$ between any pair of measurements to the spatial distance between them, $h$:

$$\gamma(h) = \frac{1}{2} E[\{Z(x) - Z(x + h)\}^2] \qquad \text{(eq. 1)}$$

An empirical semivariogram can be constructed from a set of geographically-explicit measurements by analysing all the possible paired observations, and a mathematical model can then be fit to provide a continuous estimation of the relationship

between any pair of points. This function can then be used to derive interpolation weights, being the basis of a family of interpolation methods known as Gaussian process regression or, in the geostatistical literature, kriging. Kriging interpolation yields best linear unbiased predictions (BLUPs) at unsampled locations, being a major reason for its widespread use.

The most frequently used form of kriging is ordinary kriging (OK), in which the interpolated values are linear weighted averages of the $n$ available observations, $z(x)$, and an unknown constant value, $Z(x_0)$:

$$\hat{z}(x) = \sum_{i=0}^{n} \lambda_i \, z(x_i) - Z(x_0) \qquad \text{(eq. 2)}$$

where $\lambda_i$ are the interpolation weights, with the condition that they sum to one ($\sum_{i=0}^{n} \lambda_i$) so the interpolation is unbiased. Although kriging does not require any distribution assumptions on the data, OK relies on second-order stationarity. That is, it is assumed that the expected value of $Z(x_0)$ is constant over the spatial domain ($E[Z(x_0)] = \mu(x) = m$); and that the covariance for any pair of observations depends only on the in the distance between them ($E[\{Z(x) - m\}\{Z(x + h) - m\}] =$

$c(h)$).

Here we used two extensions of OK, universal kriging (UK) and indicator kriging (IK). Universal kriging relaxes the first assumption and allows dealing with a spatially non-stationary mean, sometimes called a spatial trend. The interpolated values thus consist of a deterministic part (the trend) $\mu(x)$, and a stochastic part or residual, $\rho(x)$:

$$\hat{z}(x) = \mu(x) + \rho(x) = \sum_{k=0}^{m} \alpha_k \, f_k(x) + \sum_{i=0}^{n} \lambda_i \, z(x_i) \qquad \text{(eq. 3)}$$





where $f_k(x)$ are spatially-varying variables and $\alpha_k$ are unknown regression coefficients. Therefore, UK allows for the inclusion of co-variables as predictors for the interpolation, and can therefore be viewed as a mixed-effects model, or a combination of regression and interpolation.

Indicator kriging, on the other hand, is useful for binary variables (event / no event), and provides an estimation of the transition probability. It uses an indicator function to transform the variable into a binary outcome instead of working with

the original variable, yielding event probabilities as a result, $\hat{p}(x) = \hat{p}(z(x) = 0)$. IK can be based on either OK or UK, accepting co-variables as spatial predictors in the latter case.

Here we adopted a two-step approach, consisting on using IK for predicting precipitation occurrence, and UK for predicting the precipitation magnitude. This is an approach most commonly used for the interpolation of daily precipitation (Hwang et al., 2012; Serrano-Notivoli et al., 2019) and less so for monthly data. In the case of our study area, as we will see later on, the

frequency of zero-precipitation months is not irrelevant, so a two-step approach was advisable. Therefore, in a first step we used the following indicator function to transform the observed variable in mm into zero-precipitation events:

$$I(x) = \begin{cases} 1 & \text{if } z(x) = 0 \\ 0 & \text{otherwise} \end{cases} \qquad\qquad \text{(eq. 4)}$$

Then, we used indicator kriging to obtain estimated zero-precipitation probabilities, $\hat{p}(x)$. In a second step we used universal kriging for estimating precipitation magnitude, $\hat{z}(x)$. Once the two predictions were performed, we combined them into a

single estimated precipitation field $z'(S)$ according to the following rule:

$$z'(x) = \begin{cases} 0 & \text{if } \hat{p}(x) \le p_t \\ \hat{z}(x) & \text{otherwise} \end{cases} \qquad\qquad \text{(eq. 5)}$$

where $p_t \in (0, 1)$ is a classification threshold. Determining the classification threshold is a complex task, since different values can be used that lead to better performance on the event of interest (zero monthly precipitation, in our case) at the cost of allowing more false negatives, or the contrary. We shall discuss about the classification in the discussion section of this

article.

We used five co-variables for the deterministic part: the easting and northing coordinates, the altitude, the distance to the coast line (

Figure A 1), and the monthly climatology: zero-precipitation probability for IK (Figure A 2), and mean precipitation for UK (Figure A 3). To obtain the climatologies we computed spatial fields of monthly mean precipitation using UK and the

geographic co-variates mentioned above, based on data from a sample of 1698 observatories with at least 35 years of data over the period 1961-2000. This period was selected because it contained the highest number of serially-complete data series, while encompassing a long-enough period (40 years) to allow for stable average values of the two variables of interest. In the discussion section we provide a comparison of this approach with using the original data and using a full normalization. All co-variables were re-scaled to a common range between 0 and 1 to facilitate model parameter fitting.



One peculiarity of the variable of interest (monthly precipitation) is that it can only take non-negative values. Also, when a number of observations are considered over a sufficiently large area, the data often shows a skewed distribution. One common solution to both issues is to use a logarithmic transformation of the data, i.e. interpolating on $ln(x)$ instead of $x$, an approach that is sometimes known as lognormal kriging in the geostatistics literature. This generates additional issues, though, as this approach tends to over-estimate lower precipitation and under-estimate high precipitation (Roth, 1998).

Another drawback is that it does not allow to interpolate observations of zero precipitation, which is sometimes solved by adding a small offset, e.g. interpolating on $ln(x + 1)$, although this has other undesired effects and it does not provide a good estimation of zero observations. Here we opted for using the original variable, i.e. without a log transformation, although we compare both options in the discussion section.

   Another transformation that is often applied when interpolating precipitation fields, especially with weighted averaging

methods, is to transform the original values to anomalies. This is in fact a way of normalizing the data in space, as it eliminates the differences that occur between locations that systematically tend to receive much higher or lower precipitation. As new observatories appear and disappear around a given point, this could lead to biases in the interpolation that could introduce anomalies in the interpolated series. Here we decided to use anomalies computed as ratios to the long-term climatologies (mean values) computed above.

Selection of the semivariogram model and parameter fitting are fundamental steps for kriging. There are many different semivariogram models available, and there is no general rule as to how to choose one over the other, but the modeller's experience. While some models offer a greater flexibility to adapt to the empirical semivariogram, parameter estimation can become a problem in some cases because there are no exact solutions and the iterative algorithms used do not always yield good results. This is usually not a problem when performing one interpolation as the analyst often tries different options and

checks that there are no substantial differences between the results, but it can be an issue when a large number of interpolations need to be done and an automatic process needs to be designed, as it was the case here. We used the functions `krige` from the *gstat* package for R (Pebesma, 2004; Gräler et al., 2016) to perform the kriging interpolations, and `autofitVariogram` from the package *automap* (Hiemstra et al., 2008) to compute the semivariogram coefficients. The Matérn semivariogram model was the most frequently selected one, both for indicator and universal kriging. It is a highly

flexible model that often yields optimum results. In some cases, though, a Gaussian model was preferred by the automated procedure, and in a few cases the automatic process was not able to converge to good parameter values, so a spherical or an exponential (less flexible but more robust) model was enforced. The frequency of each semivariogram model over the whole time period is provided, for IK and UK, in the additional material (Table A 1).

   Best linear unbiased predictions (BLUPs), characterized by their mean and standard deviation, were then cast over a point

grid at regular distance over longitude and latitude, with a mean distance of 10 km between points.



## 2.3 Evaluation

Evaluation of the interpolation results is fundamental to fully understand the benefits of the interpolated dataset, the limitations and the best use cases. Here we performed a thorough evaluation based on several statistics and checks, for both the IK (probability of zero precipitation) and the UK (precipitation magnitude) interpolations. To evaluate the performance of the interpolation method to estimate values at unmeasured locations we followed a leave-one-out cross-validation (LOOCV) approach. This is an iterative process in which the interpolation is repeated as many times as there are data, each time removing one observation from the training data set that is later used to compare the estimated and observed values. Although this is a time-consuming process, it allows to obtain an independent sample that better represents the ability of the model to estimate values when no data are available. By not removing other observations that the one being used for evaluation, we could also test the effect of having a varying number of observations in the vicinity.

Validation of indicator kriging (probability of zero-precipitation). There is no consensus about the most appropriate statistic to evaluate binary classifications and their associated confusion matrices. A confusion matrix, also known as an error matrix, has four categories: true positives, TP (pred = 0, obs = 0); true negatives, TN (pred > 0, obs > 0); false positives, FP (pred = 0, obs > 0); and false negatives, FN (pred > 0, obs = 0); plus the total positive (P) and negative (N) observations and the positive and negative predicted totals (PP=TP+FP and PN=TN+FN, respectively). A variety of statistics can be calculated based on these quantities, of which here we focused on the following ones:

- The positive prediction value (PPV) is the fraction of positive predictions that are true positive: $PPV = PP/TP$. A high PPV can be interpreted as indicating the accuracy of such a statistic.

- The negative predictive value (NPV) is the fraction of negative predictions that are true negative: $NPV = NP/TN$.

- The true positive rate (TPR) is the fraction of positive cases correctly predicted: $TPR = TP/(TP + FN)$. It can be considered as the probability of detection (if a case is positive, the probability that it'd be predicted as such).

- The true negative rate (TNR) is the fraction of negative cases that were correctly predicted: $TNR = TN/(TN + FP)$. It can be considered as a measure of how specific is the test (if a case is negative, the probability that it'd be predicted as such).

The PPV and NPV are not intrinsic to the test as they depend also on the event's prevalence (fraction of positive cases in the observed sample). In a highly un-balanced sample, such as the case of zero precipitation in our dataset were the proportion of station / months with zero precipitation is very low, these two statistics will be highly affected. The TPR and TNR, on the contrary, do not depend on prevalence so they are intrinsic to the test. In diagnostic testing the TPR and TNR are the most used, and are known as sensitivity and specificity, respectively. In informational retrieval the main ratios are the PPV and TPV, where they are known as precision and recall.

We also computed two metrics that summarize the elements of the confusion matrix, so they can be considered as overall measures of the quality of the binary classification. The F1 score is computed as:



$$F1 \ = \ \frac{2\,TP}{2\,TP+FN+FP} \qquad\qquad \text{(eq. 6)}$$

As it can be seen, the F1 ignores the count of true negatives, so it places more emphasis on the positive cases (zero-precipitation months, in our case). The Matthews correlation coefficient (MCC), on the other hand, produces a high score only when the prediction results are good in all the four confusion matrix categories. It is equivalent to chi-square statistics for 2 x 2 contingency tables. Its value ranges between -1 and 1, with values close to zero meaning a bad performance (not higher than a random classifier), while 1 represents a perfect classification.

$$MCC = \frac{(TP\times TN)-(FP\times FN)}{\sqrt{(TP+FP)(TP+FN)(TN+FP)(TN+FN)}} \qquad\qquad \text{(eq. 7)}$$

For the evaluation of the magnitude we used the following metrics:

- The mean absolute error (MAE) and the relative MAE (RMAE), as global error measures:
- The mean error (ME) and the relative ME (RME), as global bias measures:
- The ratio of standard deviations (RSD), as a measure of variance deflation:

- The Kling-Gupta efficiency (KGE), as an overall goodness of fit measure.

Model evaluation was performed globally considering the whole dataset, but also for each month individually to make it possible to analyse the temporal evolution of the performance statistics.

Finally, in order to determine the benefits of the data rescue process, we compared the original (BNDC) and augmented (BNDC+YB) datasets. We used the same cross-validation scheme described above, but in this case the validation was

restricted to the period covered by the yearbooks (1916-1950). Another important difference is that we only used the observations present in both the BNDC and BNDC+YB data sets for computing cross-validation statistics, although the whole data sets were used for performing the interpolation.

## 3 Results

### 3.1 Data rescue and quality control

The annual weather yearbooks proved to be an outstanding source of climate data over the 1916-1950 period, as it contained 369,286 observations from 4,248 stations, compared to 281,951 from 2,732 stations for the BNDC in the same period. As expected, there was a significant overlap between the two sources, so the augmented data set resulting from their combination contained 432,183 observations from 4,414 stations. Figure 1 shows the temporal evolution of the two data sets

and their combination over 1916-1950. With the exception of a few years (1932, 1933, and 1937-1941), the yearbooks set overpassed the BNDC in number of stations. The improvement of the combined data set with respect to the BNDC ranged between more than 100% before 1920 and between 80 and 20% in the remaining of the period.



A striking characteristic of the dataset during this first period was the abundance of short-lived time series, and even more so
in the yearbooks data (Table 2). The highest frequency (43%) corresponds to series with less than five years of data, while
65% of the series had no more than 10 years. On the other hand, less than 5% of stations cover the complete period 1916-
1950.

The complete dataset, spanning 1916-2020, contained more than 3.3 million records from 11,008 weather stations. The
number of stations currently active in any given year was much lower, thought, and it varied significantly (Figure 2). After a
first decade (1916-1925) with no large variation at around 800 active stations, the number of active stations increased
steadily, reaching approximately 2500 stations in 1950, and peaking at 5,237 in 1975. The number of active stations has
progressively decreased since, reaching 2,615 in 2020. The most notable exception to this general trend was the Spanish
Civil War period between 1936 and 1939 and the immediate post-war years, when the dataset was severely reduced and
reached its lowest count at 675 active stations in 1939.

As a consequence of this variation, the information spatial density has greatly changed over time, too (Figure 3). Also, and in
particular in the first half of the 20th century, the spatial coverage of the dataset is not homogeneous as some regions have a
notably lower information density. Regarding the spatial coverage, the image illustrates that the addition of the rescued data
(YB dataset) improved in a significant way the data density in some regions that were severely under-represented in the
original data set (BNDC), such as the South-West (Guadalquivir River basin).

As temporal changes in data availability can have an impact in the interpolation results and in any ulterior analysis, we
assessed it in more depth. Figure 4 informs on the evolution of the mean distance to the closest neighbouring station. Since
the random component of kriging is essentially a distance-based weighting scheme, this is a relevant statistic that is related to
the degree of spatial smoothing introduced by the interpolation. Prior to 1940 the mean distance ranged between 10 and 12
km, rapidly decaying to less than 6 km after 1960. The minimum average distance (5.9 km) was achieved in 1973, with a
slight increase since then up to the present. Interestingly, the increase of the closest neighbour distance in recent years has
been slower than the reduction in the number of observatories, evidencing a more even spatial distribution of the observation
network that is also apparent when the spatial distribution of the stations in 2015 and 1955 are compared. In any case, the
strong variability of the number and density of observations during the study period is a potential source of undesired effects
in the interpolated dataset, reinforcing the need for a thorough validation.


Another potential source of bias arises from the altitudinal distribution of the stations, since there is usually a good
correlation between precipitation and elevation, so ideally the observations should sample evenly all the altitudinal ranges on
the study area. In our case, the areas below 500 m a.s.l. tended to be over represented when the proportion of observations
per altitude ranges was compared to that of the study area, while higher areas were slightly under-represented (Figure 5).
Strong temporal changes in the altitudinal distribution of the stations could be an additional problem as it could generate
temporal bias in the resulting grids. However, in our case the altitudinal distribution of the observations changed only
slightly, with the average elevation oscillating between 575 (1st quartile) and 614 (3rd quartile) m a.s.l., values that





correspond approximately to the mean elevation of mainland Spain (Figure 6). The relative composition of the dataset by altitudinal classes has not changed significantly over the study period, so no temporal bias was to be expected due to elevation shifts (Figure A 4).

Regarding the length of the data series, the frequency of observatories with less than five years of data is 22%. 34% of the series cover more than 30 years, while only a few (0.1%) cover the complete period (Table 3).

While this heterogeneity of record lengths is not uncommon in observational datasets, it imposes an important decision that conditions the development of gridded dataset: whether to use all the information available at any given moment, even if the data availability changes over time, or to restrict the analysis to a reduced set of stations that do not change over time. The last option implies selecting the largest and most complete data series and then undergoing a gap-filling and reconstruction process so in order to make all the series cover the whole period of study, at the cost of rejecting a large amount of valid data and the risk of introducing statistical artifacts during the reconstruction process. The first option, on the other hand, has the advantage of making use of all the information available, but the risk of introducing statistical biases in the dataset since the number of observations change largely over time. We shall discuss this issue later in the article.

Figure 7 shows the time series of anomalous data discarded by the quality control process. Annual anomalous data does not present any trend over time, nevertheless consider as percentage of total data discarded data during they decreased noticeably particularly before 1935, remaining in a very low percentage along the time.

These data are distributed globally on a monthly basis, as shown in Table 4, where a lower prevalence of errors can be seen in the months of July and August than during the rest of the year. This happens because during these months the volume of precipitation is lower and therefore it is more complex to detect anomalies.

## 3.2 Gridded dataset

**The result of the interpolation process was a gridded database of mean and standard deviation fields of the best linear unbiased predictions (BLUPS) of monthly precipitation between January 1916 and December 2020 (1272 time steps). As an**



**example of the dataset, mean and standard deviation fields are show for April 1916 and 1975 (**



Figure 8). The figures illustrate the probabilistic nature of the Gaussian process interpolation, as the mean and standard deviation fully describe the probability distribution of estimated precipitation at each point of the grid. There is a noticeable difference in the observational data density leading to each interpolated grid (as seen in Figure 3), which had an impact



mainly on the magnitude of the standard deviation field. In fact, despite a similar range of the mean predicted values, the standard deviation field value ranges were very different between the two dates, being almost double in 1916 than in 1975, revealing a higher uncertainty of the estimated values.

This is further illustrated in Figure 9 (left panel), which shows time series of estimated monthly precipitation (BLUPs) with their uncertainty levels (plus and minus one standard deviation) at four random grid points. In addition, time series of the standard deviation and of the distance to the closest observation are also provided. Two facts are apparent upon inspection of the plots: i) there is a linear relationship between the predicted precipitation and the uncertainty range, i. e. there is a larger uncertainty for higher precipitation; and ii) there is a reduction of the uncertainty range with time, which can be related to the progressive addition of new information.


### 3.3 Validation: probability of zero precipitation

The MMC statistic was used to determine the classification threshold for the interpolation of precipitation occurrence, since it provides a good balance between the prediction of positive and negative cases (Figure 10). Classification thresholds were computed for each month, and were applied globally (i.e., the same thresholds applied for the whole spatial and temporal

domains). The threshold values were lower in winter (close to 0.25) and higher in summer (close to 0.50), reflecting the seasonal variation of the prevalence of zero-precipitation. The thresholds offered a good balance as they tended to maximise the individual metrics of the confusion matrix (Figure A 5 and Figure A 6).

As a result, a reasonably good prediction of zero-precipitation cases was obtained in the summer months, when the prevalence is higher and thus more important, while during the rest of the year, when the prevalence is lower and therefore

less relevant, there was a slight under-estimation of zero-precipitation cases (Figure 11).

A detailed inspection of the evaluation statistics for the prediction of zero-precipitation (



Table 5) reveals that the interpolation was better at predicting negative cases (precipitation higher than zero) than positive cases (zero precipitation), as shown by higher TNR and NPV values over TPR and PPV. This is also reflected by the F1 score, which focuses in the ability to predict positive cases, and had values in the 0.50-0.65 range for most months. Only during the summer months (especially July and August) the skill was higher.

### 3.4 Validation: magnitude

Cross-validation results of precipitation magnitude (considering the combined result of the two interpolations by application of Equation 5) can be considered good. The probability density of the interpolated values matched quite well that of the observations (Figure 12), although the predicted values tended to be slightly more concentrated around the mode of the distribution while under-representing the lower and higher tails of the distribution. Such variance contraction is to be expected in any interpolation process, and it is more important to check for biases and temporal inconsistencies.

This can be seen in more detail when comparing the quantiles of the observed and predicted sets (Table 6). Starting by the median (50% percentile), there was a very good match between both sets, albeit a slight over-estimation can be found in most months. When considering the lower quantiles (25% and, especially, 10%) the over-estimation is more evident, while the higher quantiles (75% and, especially, 90%) show a closer match.

Cross-validation statistics for the magnitude interpolation are given in Table 7, and an example scatterplot of predicted against observed values for a 12-months period is provided in Figure 13. The MAE over the whole dataset ranged between slightly less than 7 mm in July and 17 mm in December, representing relative errors between 41% (July) and 22% (November and April). This might seem as quite high error values, but it must be kept in mind that the distribution of the variable of interest is highly skewed, so a relatively low number of very high observations contribute a lot to the statistic. In fact, visual inspection of the scatterplots in Figure 13 reveals a good match between observations and predictions, for all months.

The ME, very close to zero mm, indicates no significant bias in the predictions. The RSD was in general close to 0.9, which can be considered a good result and imply only a slight reduction of the spatial variance in the predicted precipitation fields. The KGE, finally, was quite good, too, with values ranging between 0.79 in May (worst case) and 0.85 in December-January (best months). In general, the validation results were better in winter, and worse during the summer months.

A very important issue when constructing gridded datasets over an extended period, as it was the case here, is to consider potential biases that may arise from the substantial change in the number of observations available at different times. Large temporal variation in the size of the observational data set can potentially impact several aspects of the interpolated grids, mostly their spatial variance, and even might be a source of bias. Here we checked for such changes by computing cross-validation statistics for each monthly grid independently and then inspecting the time series of said statistics, looking for temporal trends (Figure 14). Ideally, validation statistics should be time-stationary, although some effects are inevitable due to the changes in the size of the observational dataset.



The first and most obvious consequence of the variation in the number of observations is the effect on the overall accuracy, as expressed by the MAE. As the size of the observed dataset increased over time, the absolute error of the interpolation also decreased. A similar result would be obtained by inspecting the evolution of other goodness-of-fit statistic, such as the $R^2$, and is an inevitable consequence of having more data to interpolate. In our case, the reduction of the MAE was approximately two-fold, that is the error was two times higher during the first decades of the 20th century, when the
observational dataset was scarcest, than during the last decades of the study period.

More relevant than the absolute error is the evolution of the mean error, as it informs about possible systematic temporal biases that could affect, for instance, the computation of temporal trends using the interpolated grid. In principle, the unbiasedness of the kriging interpolation is independent of the size of the dataset, so no temporal bias should be expected. However, other factors related to the normalization of the data or other steps of the process could introduce undesired

effects. In our case, the ME was stationary or only exhibited very limited temporal trend, with close-to-zero values and mostly random oscillations. Only for some months (April and July being the most conspicuous) a slight increasing trend of the ME was apparent, albeit the magnitude of the difference between the start and the end of the study period (less than 0.5 mm) was very low in comparison with the magnitude of the variable.

Another well-known effect of the sample size is that it related to the variance shrinkage of the interpolation. This can be
inspected by the RSD statistic, which showed an increasing trend as the size of the available data increased. The magnitude of the difference between the start and the end of the study period ranged around 0.1, indicating that later grids had larger variability (and much closer to that of the observed sample) than earlier grids. Despite the not-too high magnitude of the effect, it is something that should be considered, for instance, if the interpolated grid was to be used for assessing variability or extreme values changes over time.

As a final result, indicative of the overall goodness-of-fit of the interpolation and considering both the error, the bias and the variability, the KGE statistic showed a steady increase during the study period ranging between values in the 0.65 - 0.7 range at the beginning of the period and close to 0.9 at the end.

A spatial evaluation of the quality of the interpolation, focusing on the KGE statistic, shows that the worst results were obtained in the summer months, and towards the South of the study area (Figure 15).


### 3.5 Evaluation of the combined dataset

The addition of new observational data digitised from the year books improved the prediction of zero precipitation and precipitation magnitude in all months, as shown by the cross-validation statistics computed over the period 1916-1950 (Table 8). The most notable improvement of the augmented data set (BNDC+YB) over the original one (GNDC) was the
stabilisation of the mean error during the first decades of the study period, which exhibited large variability in the original dataset (Figure 16).





## 4 Discussion

In the following paragraphs we discuss various aspects of our spatial interpolation approach and evaluate the performance of
alternative model choices. We used a geostatistical approach, universal kriging (also known as Gaussian process regression),
over other well-known and used approaches such as global or local regression, weighted averaging methods or splines, due a
number of reasons. On the one hand, and similar to other regression methods, kriging performs a probabilistic prediction, as
it allows obtaining not only best predictions at unsampled locations but also their standard deviation, allowing to determine
uncertainty ranges. Under appropriate assumptions, kriging yields best linear unbiased predictions (BLUPs), unlike other
weighted averaging methods that do not guarantee unbiasedness. Standard regression methods, on the other hand, only
consider fixed effects and result in best linear unbiased estimations, ignoring the random part. In a preliminary phase we
found that kriging resulted in better cross-validation statistics than, for instance, angular-weighting interpolation.

In order to make the best use of the data, we used all the observations available at each time step. As a result, the
interpolation sample varied largely on time, as the number of weather stations available was five times higher at their peak in
the middle 1970's than at the beginning on the period (1916-1940). Such a strong variation in the observational dataset is not
uncommon when analysing large temporal periods, and may have non-desired effects on the interpolated data set, advising
for a thorough temporal validation. It is evident that a bigger sample would result in reduced prediction uncertainty, but
should not result in systematic bias. We found that to be correct, as only the MAE but not the ME was affected by temporal
changes in the sample size. This implies that the interpolated dataset can be safely used for climatological analyses involving
the calculation of means or trends in the mean values, over the whole temporal range or overt shorter time spans. However,
some unexpected temporal bias was found related to certain variable transformations, which were discarded (more on that
later). Also, as it was expected, smaller samples resulted in a reduced variance (as shown by the RSD). As a result, caution is
recommended when using the data set for climatological assessments of spatial or temporal variability, extremes or quantiles
other than the mean.

A common concern that is often expressed against using a time-varying sample for interpolation is that it might introduce
biases (inhomogeneities) in the predicted time series at given locations as new weather stations appear (or disappear) in the
vicinity of the point, due to possible systematic differences between the two points. Although this is more problematic with
weighted averaging methods than with regression or kriging approaches, we decided to use a variable transformation in order
to eliminate such differences. Therefore, we transformed the original data in millimetres into anomalies (ratios to the point's
long-term climatology). Although this is not a strict requirement of kriging, interpolating the anomalies and transforming
back to the original units resulted in slightly better cross-validation results (Table A 3), and helped ensuring the statistical
continuity of the predicted time series at any given point of the grid as new observations (weather stations) appeared in the
vicinity of the point.

**We tried other variable transformations than the ratios to the climatology, with less good results. One of the most promising**
**approaches was performing a full standardization of the variable by converting the original values into standardized variates.**



**In fact, converting the observed values into Standardised Precipitation Index anomalies improved slightly the error statistics (MAE), albeit it yielded worse bias statistics (ME) and overall accuracy (KGE;**

Table A 4). The worse ME of the full standardisation might be a result of the transformation of the variable, but the most preoccupying effect was that it introduced a strong temporal component in the ME (Figure A 8), with a bias magnitude that

could no longer be considered irrelevant as it could appreciably alter, for instance, the computation of temporal trends based on the interpolated grid.

**Another important aspect of our approach is that it consisted on two steps, where the final precipitation prediction is the result of independent estimation of the probability of zero-precipitation and precipitation magnitude. This allowed attaining a better representation of zero-precipitation areas, which was especially relevant for the summer months. As a comparison, a**
**single-step approach (that is, direct interpolation of precipitation magnitude) resulted in a severe underestimation of zero-precipitation: if the prevalence of zero-precipitation cases was 8.24% in the observational dataset, using a single-step approach this value got reduced to 1.64%. Our two-step approach, on the other hand, yielded a much closer estimation at 8.07%. Underestimation was especially important during the summer, when the prevalence of zero-precipitation months is higher (Figure A 7). Using a single-step approach did not have such a remarkable impact on the prediction of precipitation magnitude,**
**leading to similar or marginally poorer cross-validation statistics (**

Table A 2.). This came as no surprise due to the low contribution of low-precipitation values to validation statistics in general, and highlights the importance of performing a thorough validation of the interpolation results that goes beyond the mere computation of error (deviation) statistics and considers other important aspects of the data, such as the prediction of zero-precipitation.

We also checked the added information of using co-variates, i.e. using a universal kriging approach, against a simpler ordinary kriging with no co-variates. We found that the co-variates resulted in better cross-validation statistics, both for the probability of zero-precipitation and for magnitude, although the magnitude of the difference was not too big (Table A 5).

Our variable of interest, precipitation, can only take positive values (once zero-precipitation has been ruled out), and it's distribution is typically skewed. I order to deal with these characteristics in a regression context, usually a logarithmic
transformation of the variable is advised, or using a logarithmic link function. However, this implies that the method's unbiasedness properties might not apply to the original variable under certain circumstances, recommending caution (Roth, 1998). Here we found that applying a log-transformation to the data yielded slightly worse cross-validation results (



Table A 6) and, similarly to applying full-standardization, it introduced a temporal bias in the mean error. Therefore, we opted to not using this transformation.

Our approach has a number of drawbacks and potential improvements. The kriging properties rely on a proper estimation of the semivariogram model, which needs to be estimated for each time step. We found that under certain circumstances the automatically derived semivariograms were flawed (either the parameter search did not converge, or the parameters were too low or too high), so we had to put extra care in designing automated checks and solutions, as described in the methodology. Also, we found that under certain circumstances the method could be too sensitive to outlier observations.

Another important limitation is the kriging assumption of spatial stationarity, as the semivariogram model is supposed to be valid across the whole spatial domain. This is clearly a sub-optimal approach for climate variables with often complex spatial behaviour such as abrupt changes and variations in the correlation range, spatial anisotropies, etcetera. One possible solution, not explored here, is the implementation of deep Gaussian process (deepGP) regression. Unlike 'shallow' kriging, as used here, deepGP introduces more than one layer of Gaussian processes and therefore allows for spatial non-stationarities

to be modelled (Damianou & Lawrence, 2012), providing a promising method for the interpolation of climate variables. Another drawback of the our approach is that, as only the information of the month being interpolated is used, a good wealth of useful information is not used. Spatio-temporal variogram models have been proposed to leverage on the self-correlation properties of climatic variables (Sherman, 2011; Gräler, 2016), especially over short time periods, but other possible approaches include the use of principal components fields, weather types or k-means field classification as co-variates for

universal kriging. All these approaches merit exploration and will be subject of future work.

**5 Data availability**. The MOPREDAScentury (Monthly Precipitation Dataset of Spain) gridded dataset can be accessed at the project's website at https://clices.unizar.es, and has been reposited with permanent handle https://doi.org/10.20350/digitalCSIC/15136. It is distributed under the Open Data Commons Attribution (ODC-BY) license,

and can be cited as Beguería et al. (2023).

**6 Conclusions**

We created a century-long (1916-2020) dataset of monthly precipitation over mainland Spain to serve as a basis for further climatologic analysis. To achieve that, we first augmented the current observational information in the Spanish National Climate Data Bank with new data digitised from climatic year books during the period 1916-1950. This allowed to almost

double the information available in the first decades of the 20$^{th}$ century, a crucial task due to the general data scarcity during that period, especially over certain regions such as the north- and south-west of the study area. The new data helped reduce the uncertainty of the interpolated dataset, and stabilised its mean error. We further used a two-step kriging method to interpolate monthly precipitation fields (grids) based on all the data available in the observational record. Each month was



interpolated independently, i.e. no information from the previous or posterior months was used besides the computed

climatology that was used as a co-variable. Other co-variables were the spatial coordinates, the elevation and the distance to the sea. The raw data in millimetres were converted to anomalies (ratios to the long-term monthly climatology) prior to interpolation. The main advantages of our approach were: i) relatively fast computation of the model's coefficients and predictions, especially compared to machine learning methods; ii) provides best linear unbiased predictions, unlike other methods such as global or local regression (which provide estimations, i.e. considering only the fixed effects and not the

random component), splines or weighted means (which do not consider co-variables and do not guarantee lowest error or unbiasedness); iii) has a probabilistic nature, allowing to estimate uncertainty ranges. A thorough cross-validation of the resulting gridded data set revealed a good estimation of precipitation values at unmeasured locations, with a slight over-estimation of low values and under-estimation of high values. No systematic biases were found, especially along the temporal dimension. The effects of the strong variation in the sample size due to changes in the observational network were

only apparent in the uncertainty of the grided predictions and in the grid spatial variability, but introduced no temporal bias. The resulting dataset is available to download with an open license. We have devised further means of improving the approach, which would be implemented in further versions of the dataset.



**Acknowledgments**

Projects CGL2017-83866-C3-3-R   (CLICES: Climate of the last Century in the Spanish mainland) and   PID2020-116860RB-C22 EXE: Extremos térmicos y pluviométricos en la España peninsular 1916-2020), funded by the Spanish Ministry of Science.

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





**Figures**

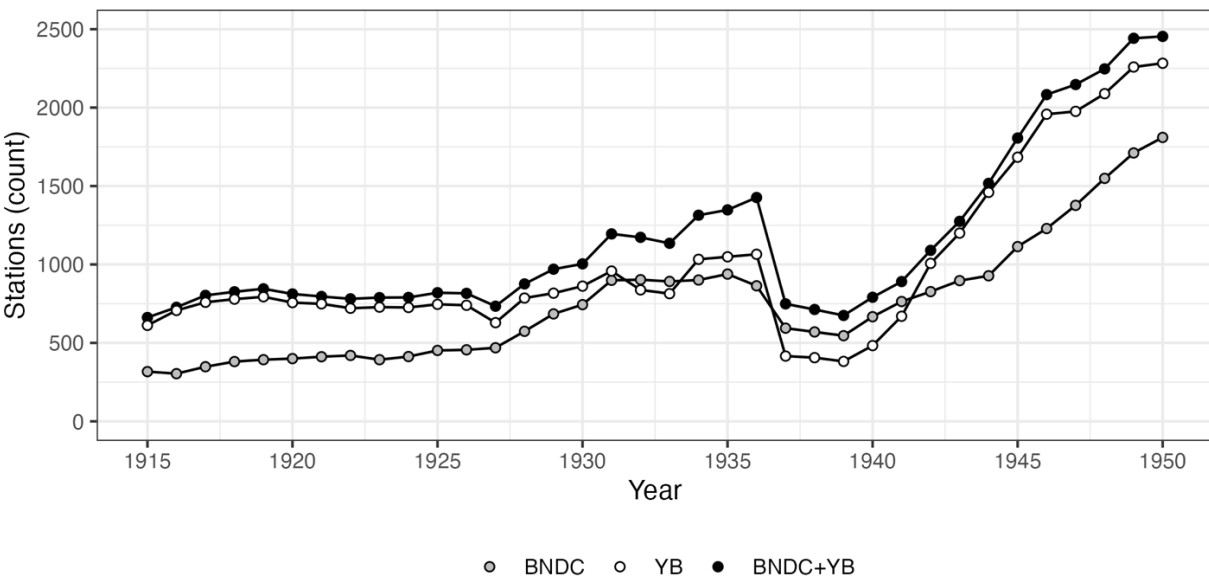


**Figure 1. Data availability (number of active weather stations per year) in the national digital data bank (BNDC) and the newly digitised year-books (YB) data sets, and their combination (BNDC+YB).**





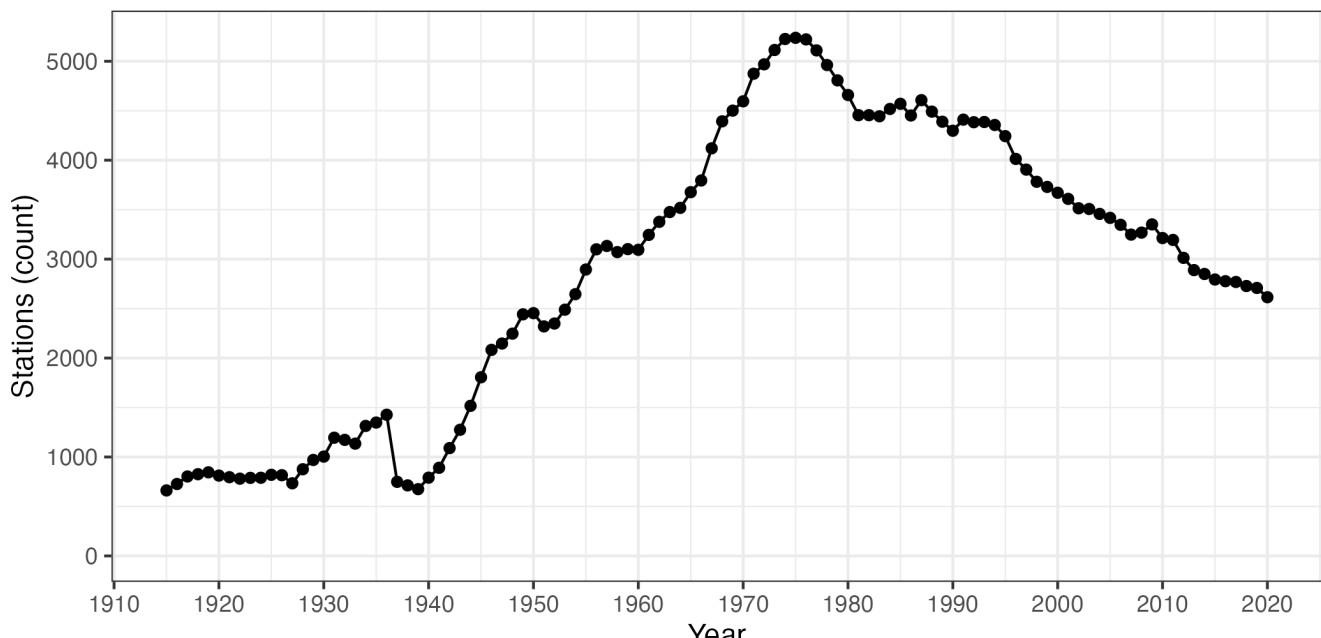

**Figure 2. Temporal evolution of the number of active weather stations in the data set over the whole study period.**

**Figure 3. Spatial distribution of the weather stations in selected years, with indication of the data origin.**



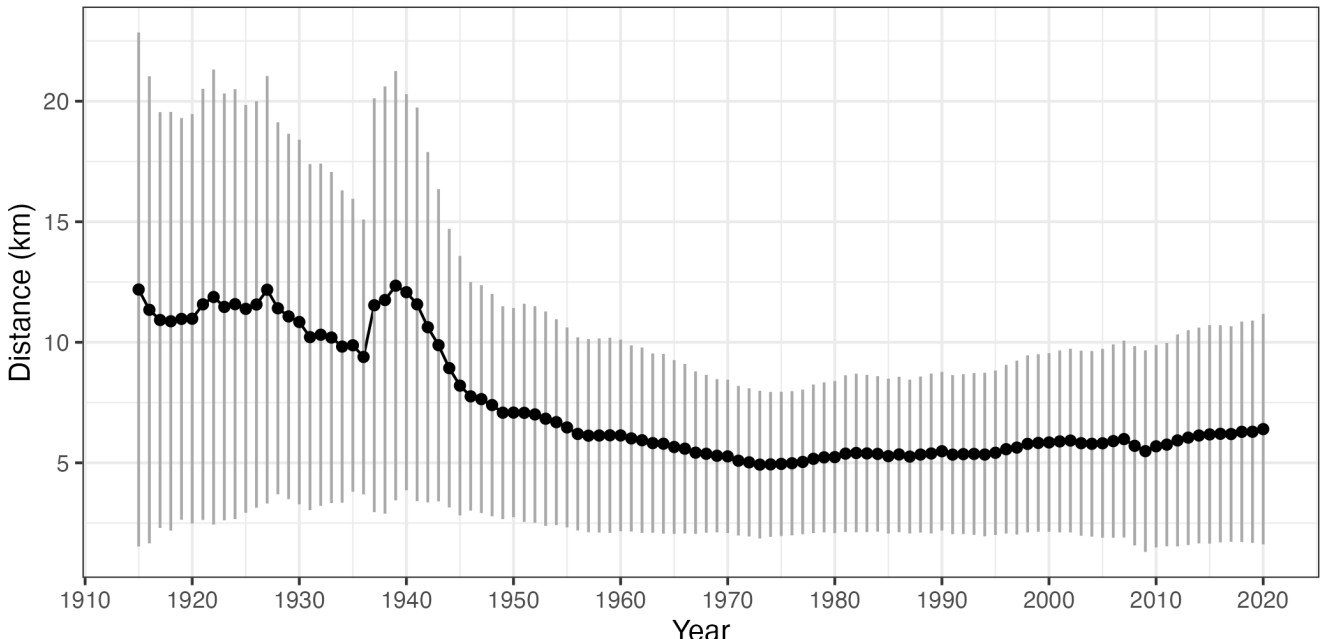

**Figure 4. Time evolution of the distance to the closest station: mean (black dots) and two standard deviation range (grey vertical lines).**





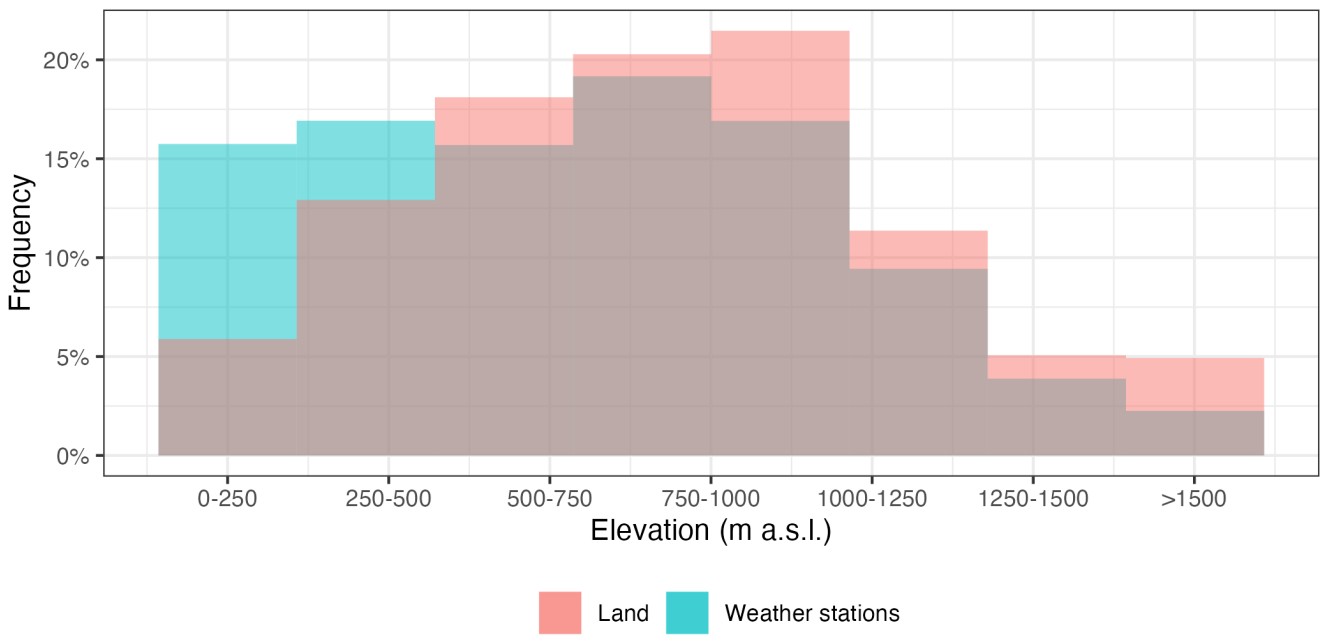

**Figure 5. Frequency histogram of weather stations per elevation class, as compared to the whole study area.**

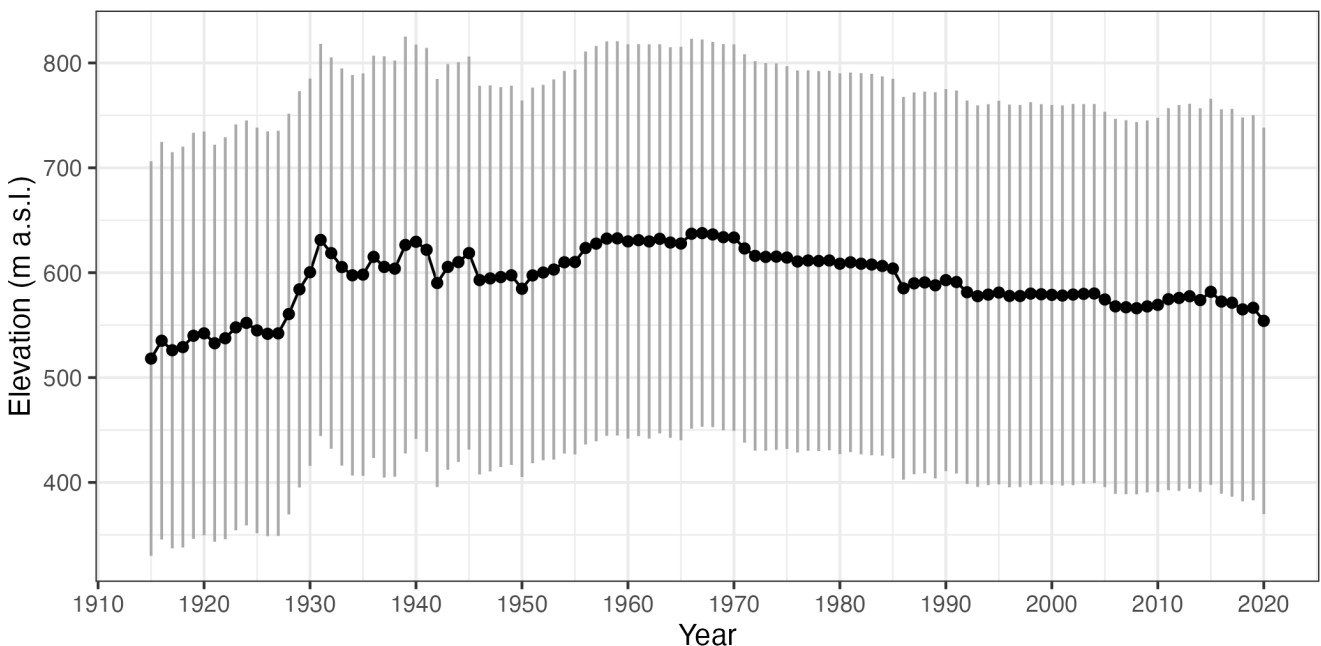

**Figure 6. Temporal evolution of the stations' elevation: mean (dots) and two standard deviation range (vertical lines).**




**Figure 7. Time series of the percent observations discarded during quality control: total number (top) and percent with respect to total observations (bottom).**






**Figure 8. Example grids: mean and standard deviation of monthly precipitation (PCP) best linear unbiased predictions (BLUPS) for April 1916 (up) and April 1975 (bottom). Black dots identify the location of four random points for further analysis.**





**Figure 9. Time series of best linear unbiased predictions (BLUPs) of April precipitation at four random grid points (left column): means (black dots) and two standard deviations range (vertical lines); standard deviation of the predictions (central column); and mean distance to the nearest observation (right column).**





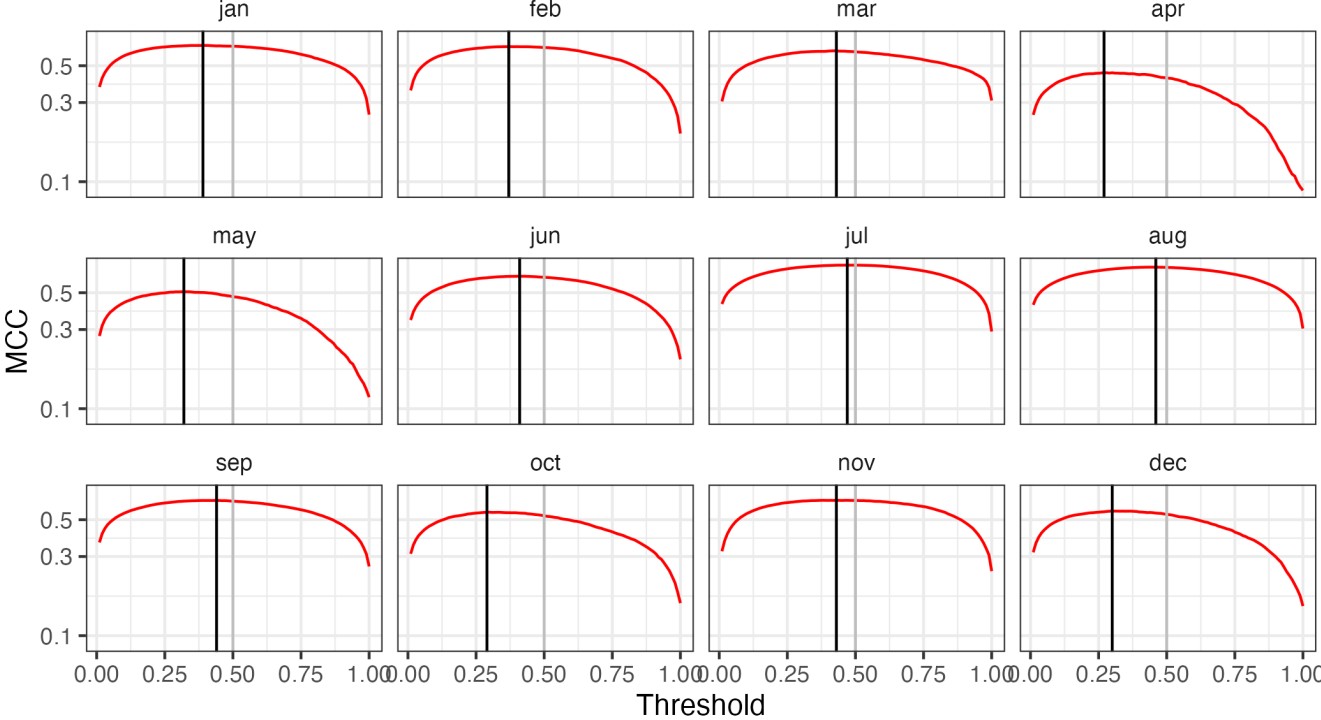

**Figure 10. Selection of zero-precipitation prediction thresholds based on the MMC statistic (cross-validation results).**

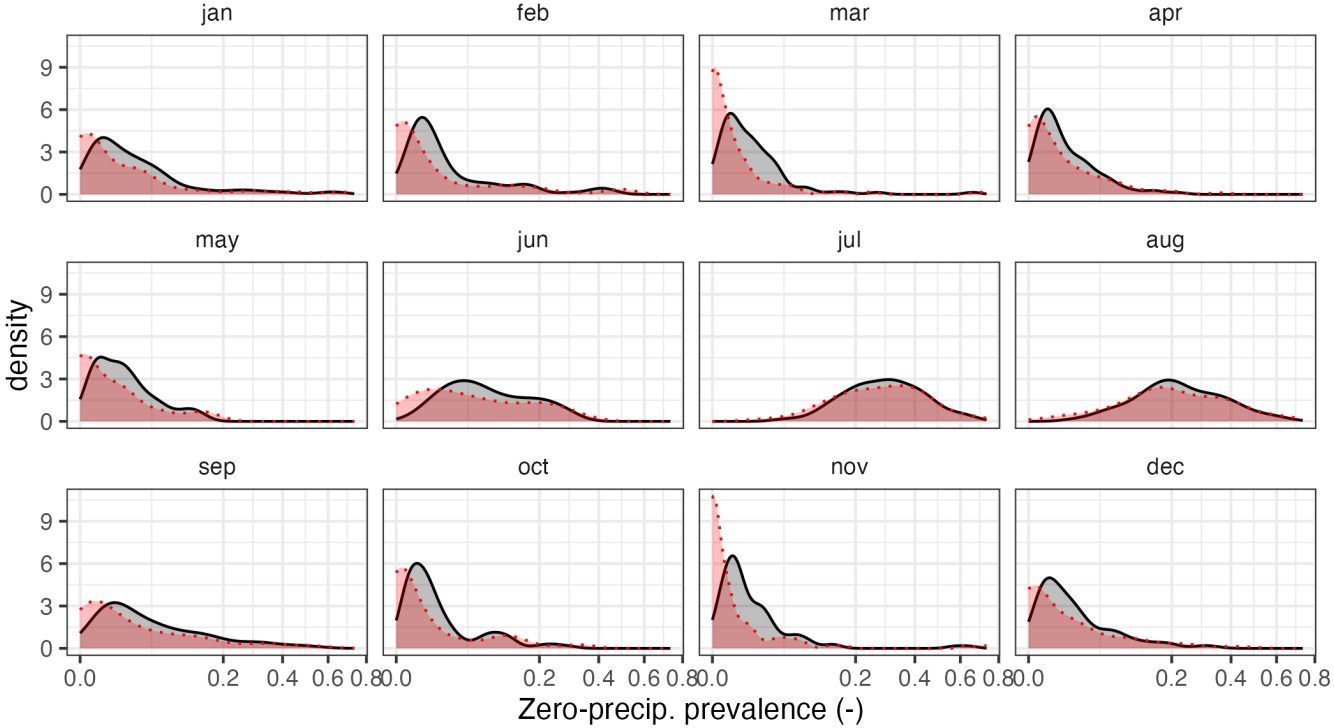

**Figure 11. Empirical density functions of zero-precipitation frequency in the observed (grey) and cross-validation (red) datasets.**





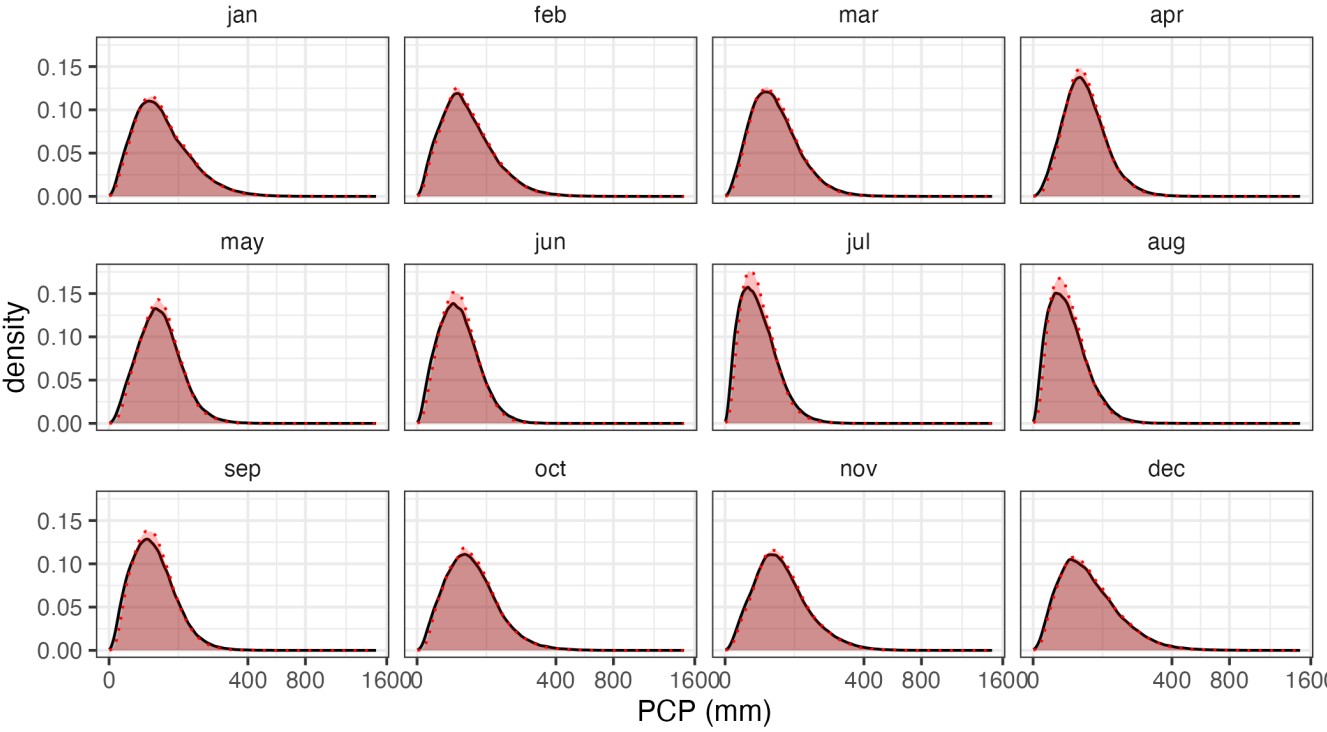


**Figure 12. Density of observed (grey) and predicted (red) monthly precipitation (cross-validation results).**

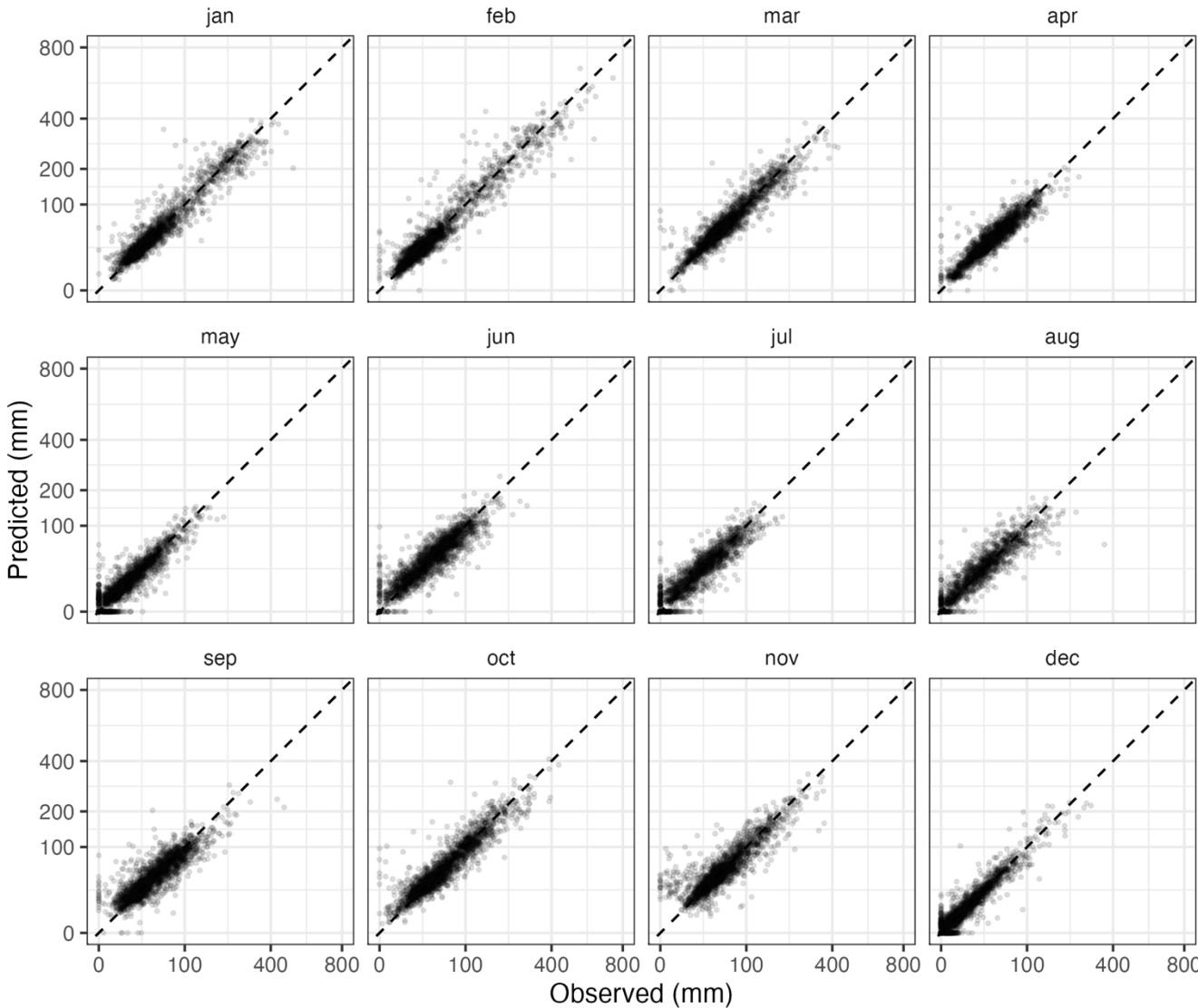

**Figure 13. Predicted and observed monthly precipitation values for year 2015 (cross-validation results). Each dot represents one weather station.**


**Figure 14. Temporal evolution of the mean absolute error (MAE), mean error (ME), ratio of standard deviations (RSD) and Kling-Gupta efficiency (KGE) for each month (cross-validation results).**


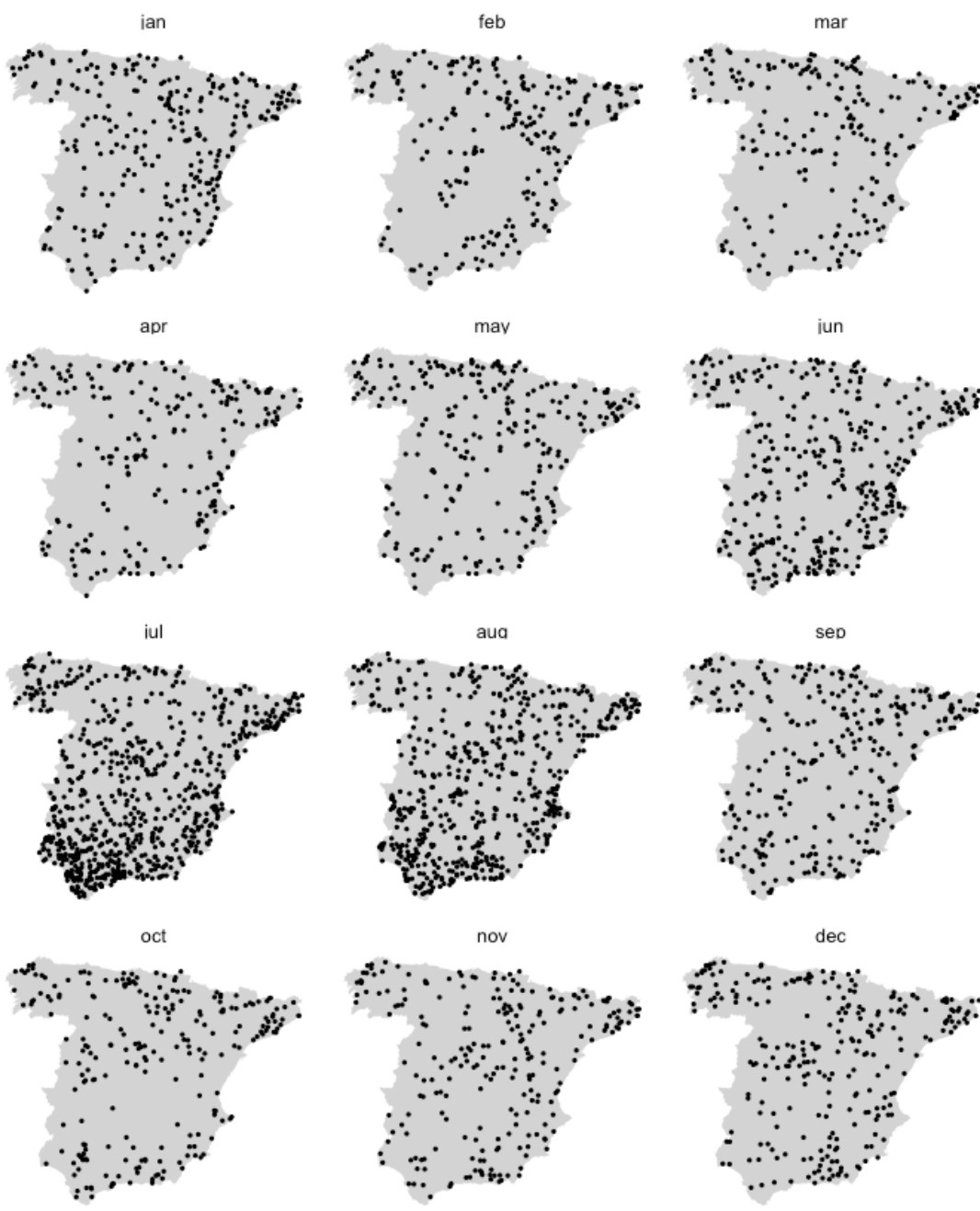

**Figure 15. Location of station / months with negative KGE over the period 1916-2020 (cross-validation results).**





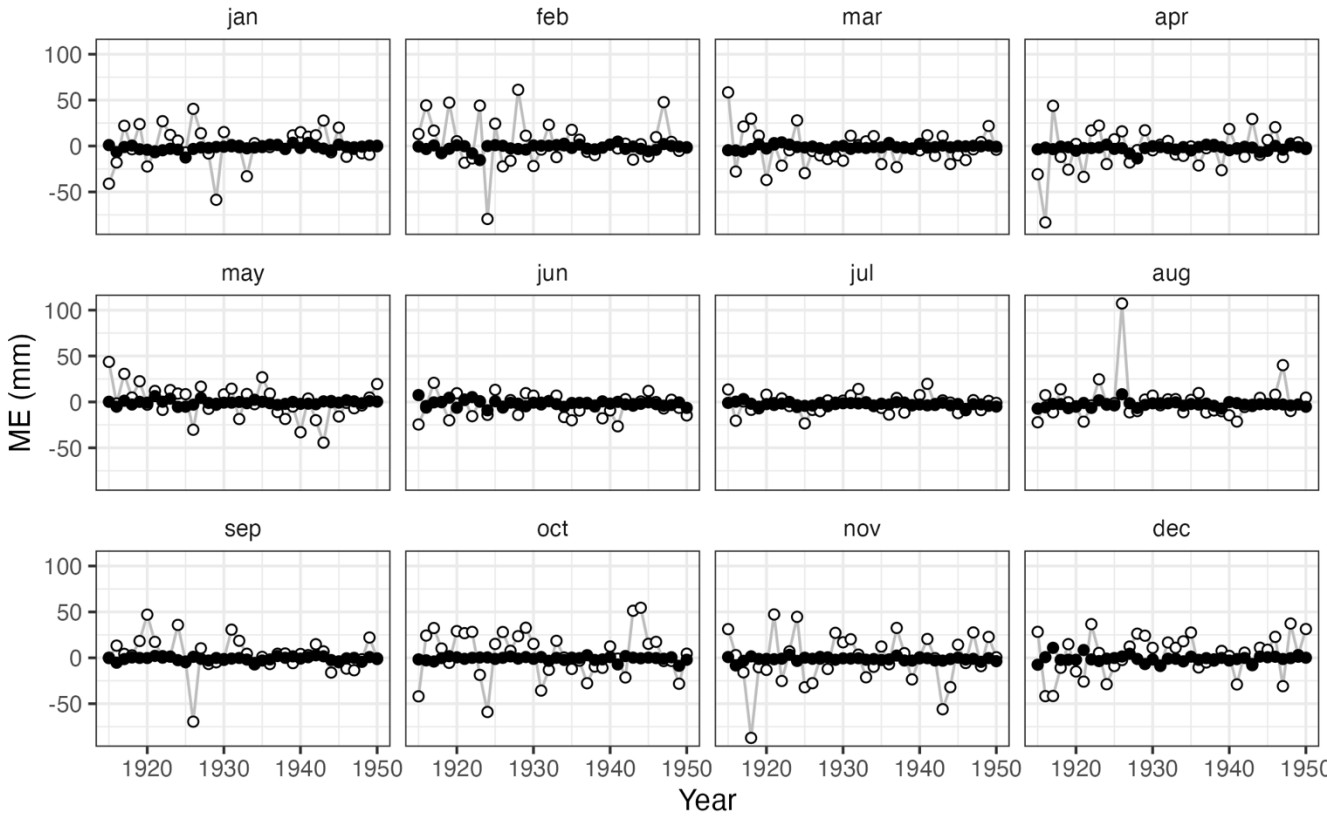

**Figure 16. Time series of the mean error in the BNDC and combined (BNDC+YB) datasets.**






**Tables**

**Table 1. Example of an anomalous value discarded by comparison with their nearest neighbours (December of 1920).**

| Suspicious data | Neighbour stations | | | |
|---|---|---|---|---|
| ID 0311A | ID 0320 | ID 0395 | ID 0390 | ID0336A |
| 6680 mm | 960 mm | 668 mm | 1332 mm | 512 mm |


**Table 2. Number and percentage of stations according to the length of the record, for the period 1916-1950.**

| Length (years) | BNDC | YB | BNDC+YB |
|---|---|---|---|
| 5 | 1095 (40%) | 1903 (45%) | 1884 (43%) |
| 10 | 660 (24%) | 1146 (27%) | 1082 (24%) |
| 15 | 334 (12%) | 541 (13%) | 524 (12%) |
| 20 | 334 (12%) | 306 (7%) | 450 (10%) |
| 25 | 124 (5%) | 156 (4%) | 221 (5%) |
| 30 | 61 (2%) | 115 (3%) | 105 (2%) |
| 35 | 78 (3%) | 68 (2%) | 98 (2%) |
| 40 | 46 (2%) | 18 (<1%) | 62 (1%) |


**Table 3. Number and fraction of stations according to the length of the record, for the period 1916-2015.**

| Length | 1 | 5 | 10 | 15 | 20 | 25 | 30 | 35 | 40 | 45 | 50 | 60 | 70 | 80 | 90 | 100 | 105 |
|---|---|---|---|---|---|---|---|---|---|---|---|---|---|---|---|---|---|
| Number | 419 | 2221 | 1244 | 1286 | 1061 | 827 | 808 | 701 | 573 | 517 | 544 | 746 | 569 | 286 | 128 | 48 | 14 |
| Percentage | 3.5% | 18.5% | 10.4% | 10.7% | 8.8% | 6.9% | 6.7% | 5.8% | 4.8% | 4.3% | 4.5% | 6.2% | 4.7% | 2.4% | 1.1% | 0.4% | 0.1% |

**Table 4. Anomalous data discarded out the total, the percentage represents distribution of the total anomalies respect the total**
**data anomalous.**

| | Jan | Feb | Mar | Apr | May | Jun | Jul | Aug | Sep | Oct | Nov | Dec |
|---|---|---|---|---|---|---|---|---|---|---|---|---|
| Anomalies | 197 | 189 | 171 | 174 | 169 | 175 | 97 | 90 | 169 | 220 | 229 | 234 |
| (%) | 9.3 | 8.9 | 8.1 | 8.2 | 8.0 | 8.3 | 4.6 | 4.3 | 8.0 | 10.4 | 10.8 | 11.0 |



**Table 5. Cross-validation statistics for zero-precipitation estimation: true positive ratio (TPR), true negative ratio (TNR), positive predictive value (PPV), negative predictive value (NPV), F1 score (F1) and Mathew's correlation coefficient (MCC). Median values across all the stations.**

| Month | TPR | TNR | PPV | NPV | F1 | MMC |
|---|---|---|---|---|---|---|
| January | 0.660 | 0.984 | 0.695 | 0.982 | 0.660 | 0.677 |
| February | 0.658 | 0.984 | 0.671 | 0.983 | 0.648 | 0.665 |
| March | 0.528 | 0.994 | 0.727 | 0.985 | 0.609 | 0.611 |
| April | 0.485 | 0.989 | 0.448 | 0.990 | 0.455 | 0.466 |
| May | 0.509 | 0.987 | 0.532 | 0.985 | 0.506 | 0.520 |
| June | 0.627 | 0.971 | 0.699 | 0.961 | 0.628 | 0.661 |
| July | 0.806 | 0.923 | 0.823 | 0.914 | 0.733 | 0.814 |
| August | 0.764 | 0.939 | 0.795 | 0.928 | 0.713 | 0.779 |
| September | 0.629 | 0.979 | 0.733 | 0.967 | 0.652 | 0.677 |
| October | 0.608 | 0.982 | 0.535 | 0.986 | 0.555 | 0.569 |
| November | 0.587 | 0.994 | 0.748 | 0.987 | 0.653 | 0.658 |
| December | 0.615 | 0.981 | 0.544 | 0.986 | 0.562 | 0.578 |

**Table 6. Mean monthly observed (obs) and predicted (pred) percentiles and mean values (cross-validation results).**

| Month | 10% obs | 10% pre | 25% obs | 25% pre | 50% obs | 50% pre | 75% obs | 75% pre | 90% obs | 90% pre | Mean obs | Mean pre |
|---|---|---|---|---|---|---|---|---|---|---|---|---|
| January | 9.94 | 12.45 | 21.17 | 24.45 | 41.35 | 43.10 | 74.60 | 73.48 | 135.20 | 132.33 | 62.34 | 62.23 |
| February | 10.65 | 13.75 | 21.60 | 24.44 | 39.48 | 41.68 | 69.94 | 69.58 | 124.30 | 120.63 | 55.80 | 55.82 |
| March | 13.31 | 16.40 | 26.39 | 29.64 | 47.80 | 50.29 | 79.67 | 78.12 | 127.54 | 122.41 | 63.86 | 63.97 |
| April | 15.90 | 19.81 | 27.85 | 30.51 | 47.12 | 48.74 | 75.22 | 74.03 | 115.04 | 109.86 | 58.46 | 58.39 |
| May | 12.50 | 16.19 | 27.00 | 31.22 | 47.30 | 49.08 | 73.36 | 72.81 | 113.37 | 109.27 | 55.04 | 55.02 |
| June | 2.10 | 4.81 | 10.39 | 13.25 | 26.45 | 29.15 | 50.00 | 49.68 | 77.33 | 73.12 | 34.57 | 34.60 |
| July | 0.00 | 0.00 | 0.00 | 0.00 | 7.10 | 9.63 | 25.20 | 25.18 | 48.04 | 44.83 | 17.64 | 17.48 |
| August | 0.00 | 0.00 | 0.90 | 3.18 | 10.95 | 12.44 | 30.19 | 30.36 | 59.11 | 55.24 | 21.48 | 21.26 |
| September | 5.81 | 9.34 | 14.71 | 18.54 | 32.27 | 35.10 | 59.17 | 57.64 | 95.62 | 89.85 | 44.89 | 44.94 |
| October | 16.05 | 18.72 | 28.82 | 31.09 | 48.55 | 50.04 | 87.44 | 87.18 | 138.85 | 134.71 | 65.91 | 65.72 |
| November | 17.20 | 20.41 | 32.70 | 35.08 | 54.77 | 57.19 | 95.01 | 94.79 | 156.67 | 152.81 | 74.66 | 74.64 |
| December | 13.07 | 16.52 | 26.90 | 29.20 | 51.55 | 52.94 | 88.20 | 87.33 | 151.39 | 150.29 | 69.99 | 69.92 |





**Table 7. Cross-validation statistics for precipitation magnitude estimation: mean absolute error (MAE, mm), mean error (ME, mm), ratio of standard deviations (RSD), Kling-Gupta efficiency (KGE). Median values across all the stations.**

| Month | MAE | ME | RSD | KGE |
|---|---|---|---|---|
| January | 11.43 | 0.60 | 0.96 | 0.817 |
| February | 10.76 | 0.59 | 0.96 | 0.814 |
| March | 10.91 | 0.57 | 0.95 | 0.812 |
| April | 11.28 | 0.55 | 0.94 | 0.818 |
| May | 11.72 | 0.48 | 0.93 | 0.806 |
| June | 9.98 | 0.22 | 0.91 | 0.762 |
| July | 6.33 | -0.08 | 0.88 | 0.679 |
| August | 6.97 | 0.00 | 0.89 | 0.703 |
| September | 10.57 | 0.25 | 0.92 | 0.794 |
| October | 12.55 | 0.50 | 0.96 | 0.838 |
| November | 12.79 | 0.60 | 0.96 | 0.83 |
| December | 13.01 | 0.62 | 0.97 | 0.83 |

**Table 8. Cross-validation statistics for zero-precipitation and magnitude in the original (BNDC) and augmented (BNDC+YB) data sets: F1 score (F1), Mathew's correlation coefficient (MCC), mean absolute error (MAE, mm), and Kling-Gupta efficiency (KGE)**
**median values across the stations, period 1916-1950.**

| | BNDC | | | | BNDC+YB | | | |
|---|---|---|---|---|---|---|---|---|
| Month | F1 | MMC | MAE | KGE | F1 | MMC | MAE | KGE |
| January | 0.975 | 0.521 | 12.080 | 0.682 | 0.974 | 0.552 | 11.663 | 0.707 |
| February | 0.973 | 0.548 | 12.886 | 0.721 | 0.975 | 0.587 | 12.380 | 0.747 |
| March | 0.985 | 0.251 | 13.993 | 0.687 | 0.991 | 0.226 | 13.187 | 0.716 |
| April | 0.983 | 0.454 | 12.379 | 0.718 | 0.980 | 0.475 | 12.101 | 0.743 |
| May | 0.986 | 0.266 | 15.722 | 0.673 | 0.989 | 0.251 | 14.882 | 0.701 |
| June | 0.950 | 0.541 | 11.496 | 0.572 | 0.952 | 0.575 | 10.826 | 0.608 |
| July | 0.907 | 0.662 | 6.540 | 0.487 | 0.904 | 0.672 | 6.292 | 0.523 |
| August | 0.926 | 0.640 | 8.725 | 0.443 | 0.917 | 0.653 | 7.970 | 0.493 |
| September | 0.973 | 0.572 | 13.938 | 0.644 | 0.971 | 0.606 | 13.338 | 0.681 |
| October | 0.985 | 0.516 | 14.100 | 0.648 | 0.984 | 0.532 | 13.360 | 0.690 |
| November | 0.985 | 0.713 | 13.737 | 0.687 | 0.985 | 0.720 | 13.417 | 0.717 |
| December | 0.985 | 0.375 | 16.252 | 0.690 | 0.985 | 0.389 | 15.243 | 0.716 |

# Appendix A: additional figures and tables


**Figure A 1. Fixed covariates used for universal kriging interpolation: easting and northing coordinates, elevation, and distance to the coast line.**








**Figure A 2. Mean monthly zero-precipitation probability over 1961-2000.**





**Figure A 3. Mean monthly precipitation over 1961-2000. All variables re-scaled between 0-1.**





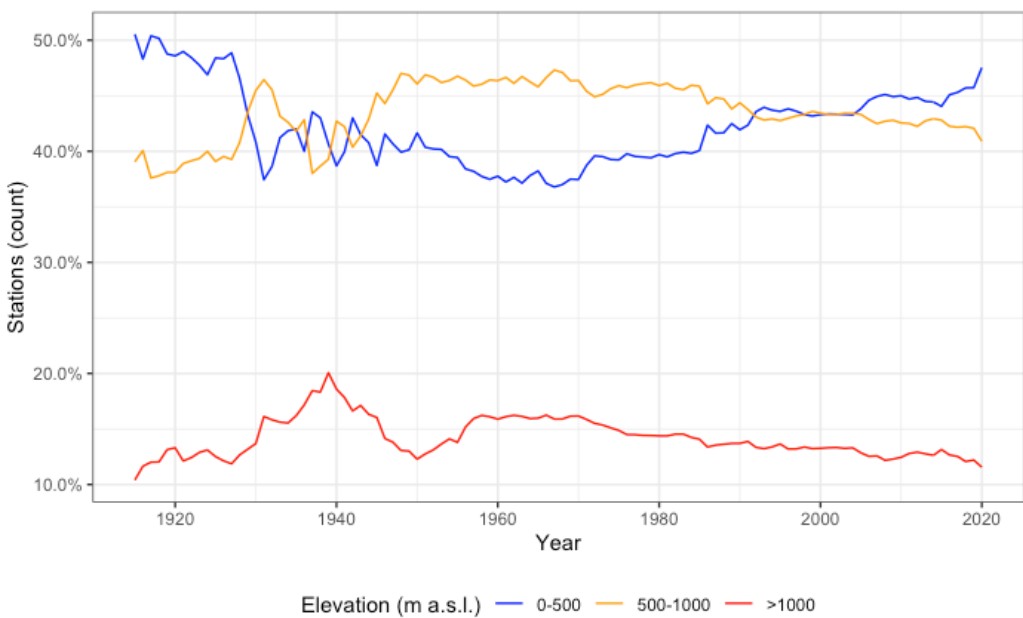

**Figure A 4. Time evolution of the frequency of observations according to altitudinal classes.**


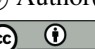


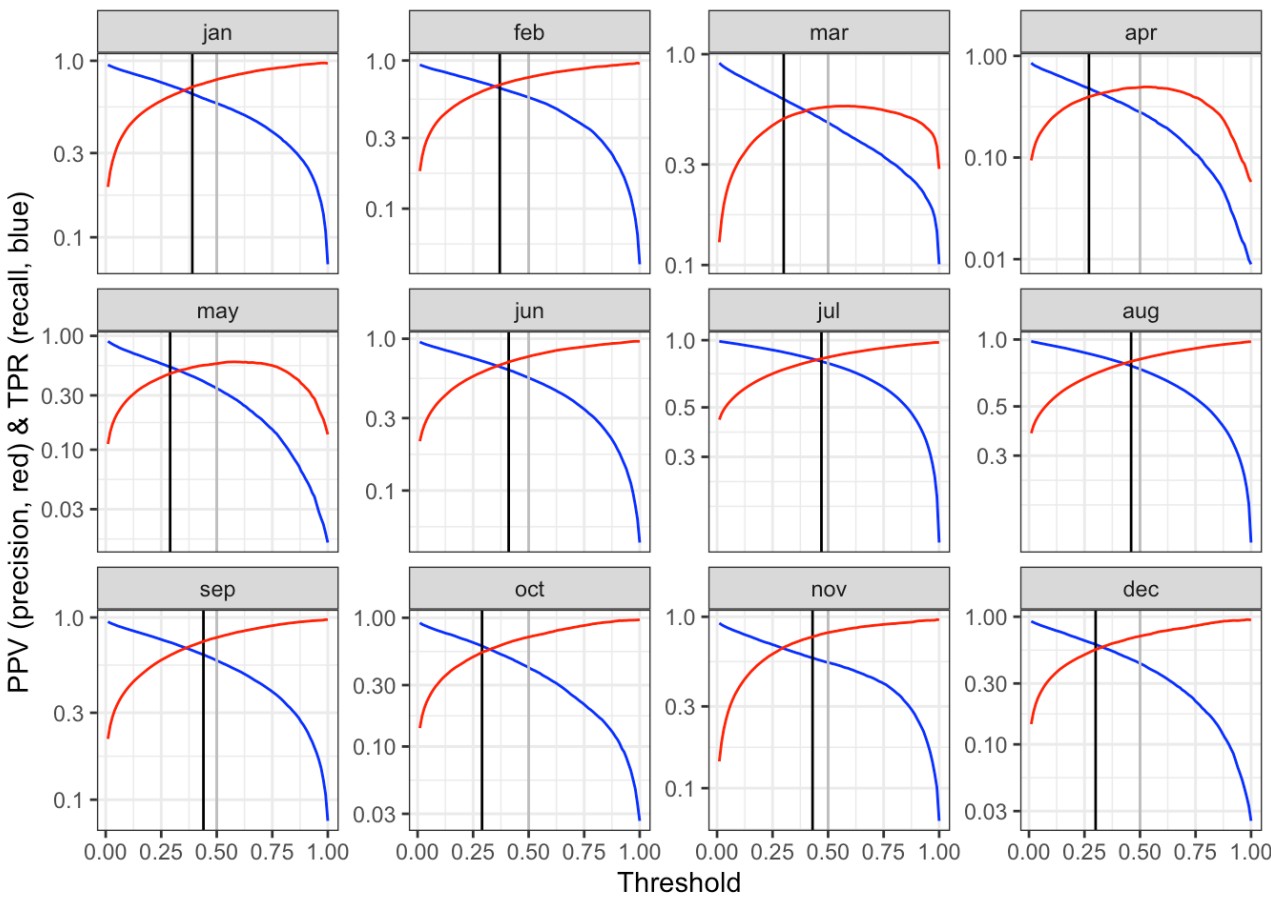

**Figure A 5. Positive prediction value and true positive rate vs. classification threshold obtained by cross-validation. The selected threshold is shown by black vertical lines.**



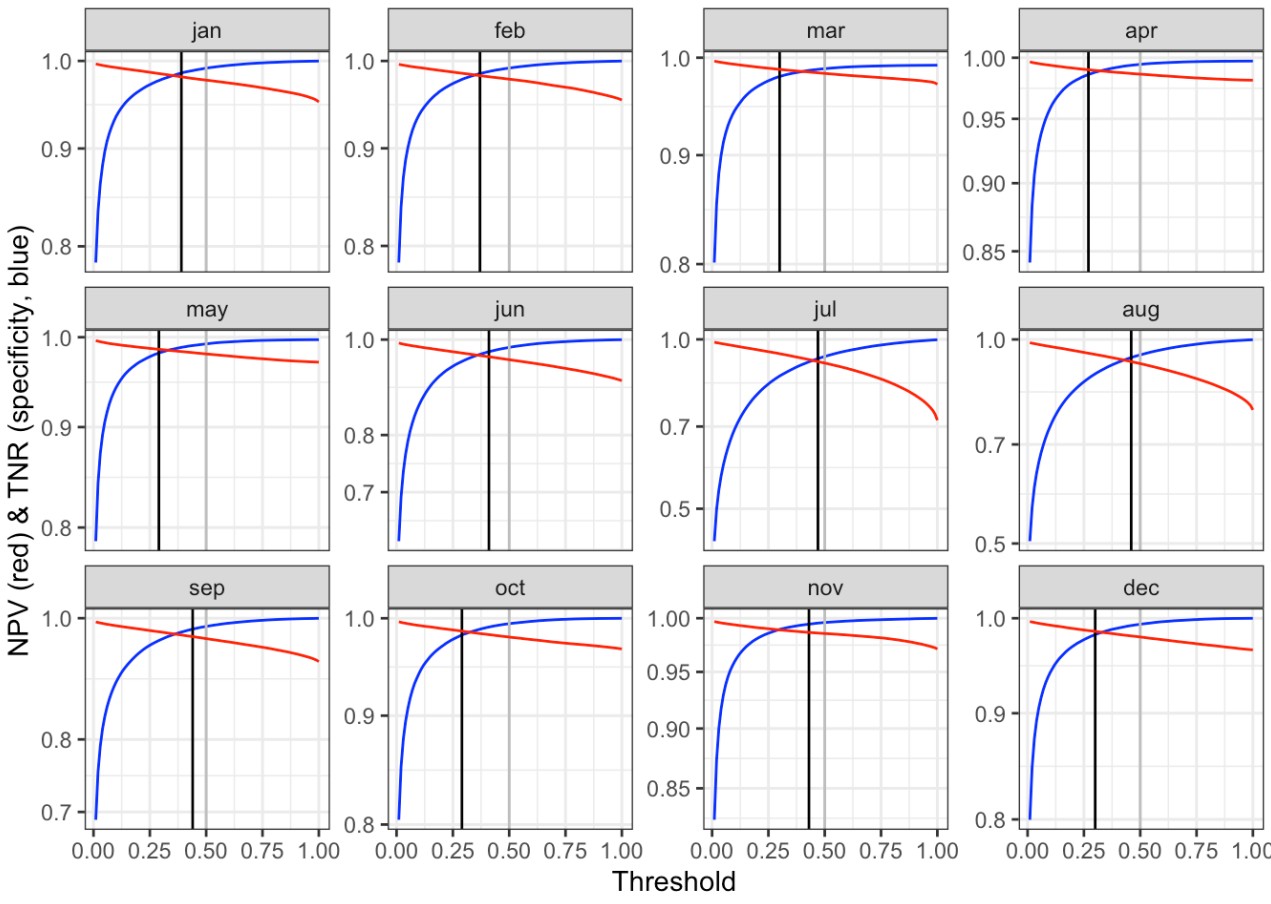

**Figure A 6. Negative prediction value and true negative rate vs. classification threshold obtained by cross-validation. The selected threshold is shown by black vertical lines.**






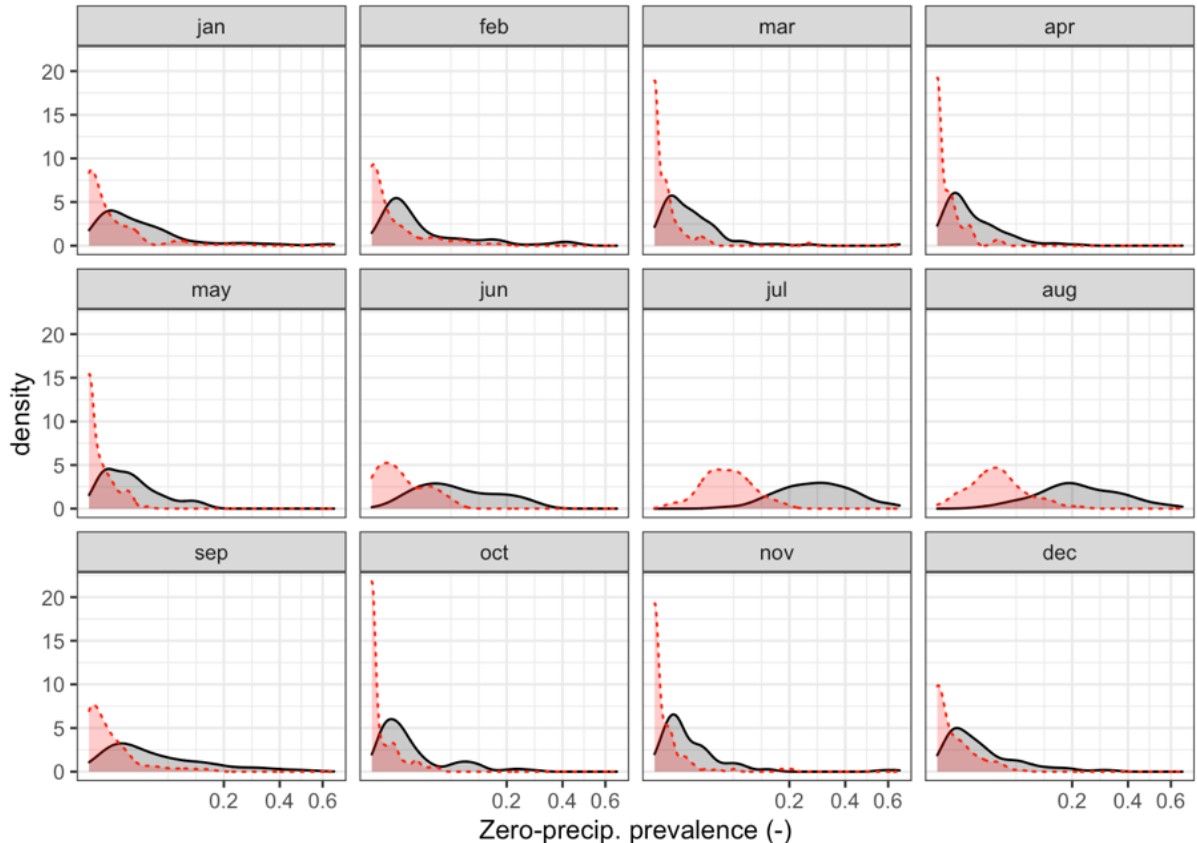

**Figure A 7. Empirical density functions of zero-precipitation frequency in the observed (grey) and cross-validation (red) datasets, using a single-step approach.**





**Figure A 8. Time series of cross-validation mean error, performing a full-standardisation of the variable.**



**Table A 1. Semivariogram models used in the interpolation of zero-precipitation probability and precipitation magnitude: exponential (Exp), Gaussian (Gau), Spherical (Sph) and Matern (Mat). Number of months.**

| | Zero-precipitation | | | | Magnitude | | |
|---|---|---|---|---|---|---|---|
| Exp | Sph | Gau | Mat | Exp | Sph | Gau | Mat |
| 39 | 454 | 44 | 735 | 35 | 89 | 11 | 1137 |

**Table A 2. Cross-validation statistics for zero-precipitation and precipitation magnitude, using a single-step approach.**

| Month | TPR | TNR | PPV | NPV | F1 | MMC | MAE | ME | RSD | KGE |
|---|---|---|---|---|---|---|---|---|---|---|
| January | 0.190 | 0.997 | 0.748 | 0.958 | 0.302 | 0.362 | 11.443 | 0.662 | 0.961 | 0.816 |
| February | 0.178 | 0.997 | 0.741 | 0.960 | 0.287 | 0.350 | 10.778 | 0.672 | 0.957 | 0.814 |
| March | 0.149 | 0.999 | 0.798 | 0.973 | 0.251 | 0.338 | 10.912 | 0.594 | 0.953 | 0.812 |
| April | 0.068 | 0.999 | 0.614 | 0.982 | 0.122 | 0.200 | 11.259 | 0.639 | 0.940 | 0.817 |
| May | 0.078 | 0.999 | 0.691 | 0.973 | 0.140 | 0.226 | 11.715 | 0.574 | 0.924 | 0.805 |
| June | 0.136 | 0.997 | 0.828 | 0.916 | 0.233 | 0.314 | 9.988 | 0.381 | 0.901 | 0.760 |
| July | 0.203 | 0.991 | 0.913 | 0.736 | 0.332 | 0.355 | 6.405 | 0.196 | 0.873 | 0.672 |
| August | 0.183 | 0.993 | 0.895 | 0.798 | 0.303 | 0.349 | 7.041 | 0.252 | 0.879 | 0.697 |
| September | 0.139 | 0.998 | 0.840 | 0.928 | 0.239 | 0.324 | 10.590 | 0.387 | 0.920 | 0.793 |
| October | 0.078 | 0.999 | 0.682 | 0.969 | 0.140 | 0.223 | 12.550 | 0.654 | 0.953 | 0.837 |
| November | 0.171 | 0.999 | 0.786 | 0.974 | 0.281 | 0.359 | 12.816 | 0.632 | 0.953 | 0.827 |
| December | 0.125 | 0.997 | 0.642 | 0.969 | 0.209 | 0.274 | 13.005 | 0.715 | 0.964 | 0.824 |

**Table A 3. Cross-validation statistics for precipitation magnitude, without variable transformation.**

| Month | MAE | ME | RSD | KGE |
|---|---|---|---|---|
| January | 11.692 | 0.599 | 0.960 | 0.813 |
| February | 10.957 | 0.547 | 0.958 | 0.812 |
| March | 11.120 | 0.581 | 0.952 | 0.809 |
| April | 11.370 | 0.538 | 0.942 | 0.817 |
| May | 11.834 | 0.475 | 0.926 | 0.805 |
| June | 10.062 | 0.222 | 0.902 | 0.758 |
| July | 6.359 | -0.085 | 0.875 | 0.678 |
| August | 7.072 | -0.023 | 0.876 | 0.698 |
| September | 10.670 | 0.248 | 0.917 | 0.791 |
| October | 12.668 | 0.471 | 0.955 | 0.836 |
| November | 13.033 | 0.558 | 0.952 | 0.824 |
| December | 13.218 | 0.542 | 0.966 | 0.822 |



**Table A 4. Cross-validation statistics for precipitation magnitude, performing a full-standardisation of the variable.**

| Month | MAE | ME | RSD | KGE |
|---|---|---|---|---|
| January | 11.317 | -0.618 | 0.948 | 0.813 |
| February | 10.675 | -0.475 | 0.946 | 0.812 |
| March | 10.818 | -0.608 | 0.941 | 0.810 |
| April | 11.152 | -0.750 | 0.934 | 0.815 |
| May | 11.757 | -0.885 | 0.917 | 0.793 |
| June | 9.843 | -1.115 | 0.887 | 0.752 |
| July | 6.088 | -0.738 | 0.850 | 0.660 |
| August | 6.917 | -0.930 | 0.854 | 0.673 |
| September | 10.350 | -1.173 | 0.908 | 0.791 |
| October | 12.453 | -0.987 | 0.943 | 0.833 |
| November | 12.738 | -0.780 | 0.941 | 0.822 |
| December | 12.884 | -0.684 | 0.953 | 0.819 |


**Table A 5. Cross-validation statistics for zero-precipitation and precipitation magnitude, interpolation with no co-variates (ordinary kriging). Median values across all the stations.**

| Month | TPR | TNR | PPV | NPV | F1 | MMC | MAE | ME | RSD | KGE |
|---|---|---|---|---|---|---|---|---|---|---|
| January | 0.679 | 0.982 | 0.668 | 0.983 | 0.655 | 0.982 | 11.619 | 0.701 | 0.959 | 0.809 |
| February | 0.658 | 0.984 | 0.671 | 0.983 | 0.648 | 0.984 | 10.943 | 0.627 | 0.957 | 0.808 |
| March | 0.552 | 0.992 | 0.687 | 0.986 | 0.604 | 0.992 | 11.021 | 0.620 | 0.950 | 0.808 |
| April | 0.441 | 0.990 | 0.467 | 0.989 | 0.444 | 0.990 | 11.419 | 0.592 | 0.938 | 0.810 |
| May | 0.517 | 0.986 | 0.522 | 0.985 | 0.505 | 0.986 | 11.835 | 0.555 | 0.921 | 0.799 |
| June | 0.626 | 0.971 | 0.694 | 0.961 | 0.626 | 0.971 | 10.013 | 0.230 | 0.898 | 0.756 |
| July | 0.800 | 0.925 | 0.828 | 0.912 | 0.733 | 0.925 | 6.288 | -0.068 | 0.872 | 0.674 |
| August | 0.773 | 0.934 | 0.783 | 0.930 | 0.710 | 0.934 | 6.987 | -0.024 | 0.881 | 0.698 |
| September | 0.627 | 0.979 | 0.733 | 0.967 | 0.652 | 0.979 | 10.593 | 0.255 | 0.919 | 0.789 |
| October | 0.546 | 0.986 | 0.577 | 0.984 | 0.547 | 0.986 | 12.753 | 0.584 | 0.952 | 0.833 |
| November | 0.591 | 0.993 | 0.735 | 0.987 | 0.649 | 0.993 | 13.048 | 0.655 | 0.951 | 0.818 |
| December | 0.575 | 0.984 | 0.566 | 0.984 | 0.554 | 0.984 | 13.230 | 0.682 | 0.964 | 0.819 |



**Table A 6. Cross-validation statistics for precipitation magnitude, using a logarithmic variable transformation.**

| Month | *MAE* | *ME* | *RSD* | *KGE* |
|---|---|---|---|---|
| January | 11.692 | 0.599 | 0.960 | 0.813 |
| February | 10.957 | 0.547 | 0.958 | 0.812 |
| March | 11.120 | 0.581 | 0.952 | 0.809 |
| April | 11.370 | 0.538 | 0.942 | 0.817 |
| May | 11.834 | 0.475 | 0.926 | 0.805 |
| June | 10.062 | 0.222 | 0.902 | 0.758 |
| July | 6.359 | -0.085 | 0.875 | 0.678 |
| August | 7.072 | -0.023 | 0.876 | 0.698 |
| September | 10.670 | 0.248 | 0.917 | 0.791 |
| October | 12.668 | 0.471 | 0.955 | 0.836 |
| November | 13.033 | 0.558 | 0.952 | 0.824 |
| December | 13.218 | 0.542 | 0.966 | 0.822 |

**Table A 7. Cross-validation statistics for precipitation magnitude, without the small offset.**

| Month | *MAE* | *ME* | *RSD* | *KGE* |
|---|---|---|---|---|
| January | 11.591 | 0.591 | 0.961 | 0.814 |
| February | 10.958 | 0.548 | 0.958 | 0.812 |
| March | 11.110 | 0.581 | 0.952 | 0.810 |
| April | 11.298 | 0.531 | 0.942 | 0.817 |
| May | 11.834 | 0.475 | 0.926 | 0.805 |
| June | 10.062 | 0.222 | 0.902 | 0.758 |
| July | 6.353 | -0.086 | 0.875 | 0.678 |
| August | 7.047 | -0.022 | 0.878 | 0.699 |
| September | 10.671 | 0.256 | 0.917 | 0.790 |
| October | 12.668 | 0.471 | 0.955 | 0.836 |
| November | 12.960 | 0.562 | 0.952 | 0.824 |
| December | 13.085 | 0.552 | 0.966 | 0.822 |