# Peer review of "MOPREDAScentury: a long-term monthly precipitation grid for the Spanish mainland"

_Earth System Science Data, 2023_

## Community Comment (CC1)

Comments to the Author

Review of essd-2023-63: 'MOPREDAScentury: a long-term monthly precipitation grid for the Spanish mainland' by Santiago Beguería, Dhais Peña-Angulo, Víctor Trullenque-Blanco, Carlos González-Hidalgo, submitted to be published in ESSD: Earth System Science Data.

Recommendation: Reject or Major Revision

**General Comments:**

This work describes the development of a new monthly precipitation dataset for the mainland Spain which is a natural continuation of the previous studies of the authors. The period covered by this dataset has not precedents in this region being of great interest to analize trends and the effect of the different uncertainty sources affecting the interpolation process on time (station's density, spatial coverage, temporal coverage, etc..). In this sense, the study is very interesting and could be a good contribution to the climatic community of the target region. However, in my opinion the work should clarify some points and/or properly discuss and support some affirmations the authors have made before to be considered for publication in ESSD.

Based on these concerns I would recommend a minor revision.

**Minor Comments:**

**Abstract:** Which is the dataset referred to by the authors? It seems to be the observational network of NCDB-AEMET but is confuse in the manuscript. Based on your summary, the interpolation process has not two phases alone, those are only referring to the dual nature of precipitation, but more, at least to obtain the monthly precipitation from the monthly anomaly and the monthly climatology.

**Introduction:** "These efforts are particularly needed in regions where water is scarce and limiting resource combined with a high demand…." I would disagree with this affirmation. In my opinion, in this sense is more relevant the spatial variability of the precipitation and the heterogeneous regimes in the region considered than the water availability as to properly characterize the precipitation in an heterogeneous area you need as much observations as possible. In particular, in addition to the water availability problem pointed out by the authors, the region considered has a great spatial variability driven by the orography. **Line 42:** What are the authors referring to with "models"? Numerical models or statistical models? In the first case, the use of gridded data is mostly due to the areal-average representativity, as the models, in contrast with the point representativity of local observations. **Line 50**: Several more recent datasets have been ignored or not included in the text. In particular, at least SAFRAN-Spain (Quintana-Seguí et al. 2016, 2017), Iberia01 (Herrera et al. 2019) and AEMET-Spain (Peral et al 2017), that have more recent periods and higher resolution than the ones cited by the authors. **Lines 57-60:** The reference for the observational datasets used have changed with respect to the abstract. Is there any reason?

**Data Rescue (yearbooks): Line 80:** These two sentences seem to be contradictory. I mean, there is a manual quality control but the the observations are automatically flagged as suspicious. There is a bold text and a reference to Table 1 but seems not being the caption of this table but normal text.

**Evaluation:** Although coherent, the notation is a litle confuse in my opinion. How do the authors define the indicator function? Usually, the 1 corresponds to the occurrence of precipitation and, then, the true positives are defined as obs > 0 and pred > 0. However, the authors use the opposite notation so I suppose that the indicator function identifies the days with 0 precipitation. This is in agreement with your equations, simply I comonly use another definition. Which is the range of the different parameters defined? I mean, PPV is lower than 1 so high values correspond to values close to 1. Similarly, the others parameters have their own range and interpretation.

**Results: Section 3.1,:** The number of stations is referred to monthly data? The nature of the data is relevant to properly interpret the added value of the yuearbooks with respect to the use only of the BNDC. It is surprising to me that, including in the modern years, the yearbooks could have this clear added value with respect the BNDC. The result about the mean distance to the closest station is reflecting that there is not any station isolated more than the dataset is spatiallu homogeneous. For example, suppose you have a dataset with 6 stations, three on the north and thre on the south of the Iberian Peninsula. If you remove one of each region, the mean distance doesn't change significantly, which is the result you obtain. I mean, is the reduction of the stations the one that is spatially homogeneous, avoiding any isolation of the stations. The effect of the stations reduction is more visible on the increment of the variability of that distance. The correlation between precipitation and orography is high when monthly of climatological values are considered but it is not so evident at shorter time scales. **Section 3.2,:** What is the meaning of the numbers inside the maps? Are they the points to which the authors refer after? The forumulation of the standard deviation of the kriging leads to the linear relation pointed out by the authors, Have they used others definitions for the uncertainty estimation? For example, the one proposed by Yamamoto (Yamamoto 2000). **Section 3.3,:** Could be the underestimation of the zero-precipitation frequency considered a common problem of gridded datasets? **Section 3.4,:** ".. variance contraction is expected in any interpolation …" instead "is to be". Could be also "should be". The parameter considered, MAE, suffers less than the RMSE of the inflation due to low number of anomalous values so the

reasoning proposed would be more clear for that index instead the MAE. The figure shown only include one year and does not explain the relative error of 41% obtained for July. How this error is in comparison with others existing datasets?

**References:**
Herrera, S., Cardoso, R. M., Soares, P. M., Espírito-Santo, F., Viterbo, P., and Gutiérrez, J. M.: Iberia01: a new gridded dataset of daily precipitation and temperatures over Iberia, Earth Syst. Sci. Data, 11, 1947–1956, https://doi.org/10.5194/essd-11-1947-2019, 2019.

Peral C.,  Navascués B., and Ramos P., Serie de precipitación diaria en rejilla con fines climáticos Nota técnica 24 de AEMET.

P. Quintana-Seguí, C. Peral, M. Turco, M.C. Llasat, E. Martin, Meteorological analysis systems in north-east Spain. Validation of SAFRAN and SPAN, Journal of Environmental Informatics, 27 (2) 116-130, 2016.DOI: 10.3808/jei.201600335.

Quintana-Seguí, P., Turco, M., Herrera, S., & Miguez-Macho, G. (2017). Validation of a new SAFRAN-based gridded precipitation product for Spain and comparisons to Spain02 and ERA-Interim. Hydrology and Earth System Sciences, 21(4), 2187–2201. https://doi.org/10.5194/hess-21-2187-2017

Yamamoto, J.K. An Alternative Measure of the Reliability of Ordinary Kriging Estimates. Mathematical Geology 32, 489–509 (2000). https://doi.org/10.1023/A:1007577916868

---

## Author Response (AR1)

**Response to https://doi.org/10.5194/essd-2023-63-RC1.**

We provide a point-by-point response (R) to the reviewers' questions (Q).

Q: "Although it is indicated that digitisation was carried out by using special flatbed scanners and manual reading and input, what are the percentages of data recovered in one way or another? And has the same quality control been applied to both?"

R: We used the scanned yearbook collection from AEMET's public repository (https://repositorio.aemet.es). The digitisation was carried out by manual reading and typing into digital files; we did not use any recognition software, at any step of the process. The digitised series were then subject to a quality control process. We have rephrased the process to make it more clear to the reader.

Q: "Digitization has been carried out on monthly data or another scale (daily, sub-daily,...)?"

R: The yearbooks provide only the monthly precipitation totals, together with other variables (such as the month's daily maximum) that were not considered for this work.

Q: "22% (43% before 1950) of the stations have 5 or less years of data (419 only 1 year). These short series usually have data quality problems. Has the contribution of these series been tested in any way, is it relevant and is it homogeneous compared to more stable series?"

R: It is not possible to test for homogeneity on such short data series, as a longer period is required in order to compute stable statistics. On the other hand, inhomogeneities are typically found in long time series due to undocumented changes in location or measuring instruments, but they are much less frequent in short-lived time series. Therefore, we included all the data provided by short-lived series, and we relied on the quality-control stage described in the text for detecting data anomalies. Given our main goal of constructing monthly fields with the highest number of data available, and not long time series at specific points, our approach is perfectly valid, in our opinion.

Q: "The quality control of digitized data is very little restrictive. Individual data that exceed certain relative or absolute thresholds are not checked, nor is visual cross-checking of the digitized data described. The type of quality control used may be sufficient to detect digitization errors that have a significant effect on the gridded product, but it does not guarantee that the retrieved series can be used with confidence for other types of climate analysis that involve the use of the individual series. An additional and exhaustive quality control on the recovered data fraction should then be done."

R: We agree that additional quality control should be done for other types of climate analysis. For our purpose of constructing a grid, however, the quality-control process described in the article proved to be sufficient to remove major artifacts that were present if the quality-control was not performed. We must say that, perhaps

surprisingly, the largest part of the data discarded during the quality control stage were not due to digitisation errors, as hypothesized by the reviewer, but were errors present in the yearbooks data.

Q: "Regarding the data discarded in the quality control, to which cases of those described in 80-85 do they correspond?"

R: The values shown in Figure 7 (original manuscript) correspond to discarded anomalous data. We have updated the figure including both anomalous data and duplicated sequences, and we have modified the text accordingly.

Q: "How many suspicious data detected at quality control have been recovered by consulting the original sources or others?"

R: The only sources consulted were the yearbooks. In a very few cases we detected errors in the digitised data base that were due to the digitisation process itself. The wrong data was therefore corrected by consulting the original source (the yearbooks). We did not keep a track of this stage so we can't quantify it, but as said it was a very rare circumstance.

Q: "In line 89 "example of data rejection is provided in..." it is not indicated."

R: The example is provided in Table 1, referenced in the text.

**Response to https://doi.org/10.5194/essd-2023-63-RC2**

We provide a point-by-point response (R) to the reviewers' questions (Q). Note that this is the same review as https://doi.org/10.5194/essd-2023-63-CC1, so our responses apply to both.

Q: "Abstract: Which is the dataset referred to by the authors? It seems to be the observational network of NCDB-AEMET but is confuse in the manuscript."

R: Yes, we refer to the National Climate Data Bank of the Spanish meteorological service (NCDB-AEMET). We have made it clearer in the new version of the manuscript.

Q: "Based on your summary, the interpolation process has not two phases alone, those are only referring to the dual nature of precipitation, but more, at least to obtain the monthly precipitation from the monthly anomaly and the monthly climatology."

R: Here we refer to the process for creating the monthly grids, which consists of two steps, or two interpolations. It is true that we also used spatial interpolation for creating the climatology grids, as it is explained in the methods section, but this was made only once for the whole process and we considered it as a pre-calculation. Therefore, we did not mention it the abstract to not make it too cumbersome. It is fully explained in the methods section, though.

Q: "Introduction: "These efforts are particularly needed in regions where water is scarce and limiting resource combined with a high demand...." I would disagree with this affirmation. In my opinion, in this sense is more relevant the spatial variability of the precipitation and the heterogeneous regimes in the region considered tan the water availability as to properly characterize the precipitation in an heterogeneous area you need as much observations as possible. In particular, in addition to the water availability problem pointed out by the authors, the region considered has a great spatial variability driven by the orography."

R: We fully agree with the comment, and we have modified the sentence to include the suggested text. Spatial variability and complexity is indeed one of the reasons for developing denser datasets, such as we did here.

Q: "Line 42: What are the authors referring to with "models"? Numerical models or statistical models? In the first case, the use of gridded data is mostly due to the areal-average representativity, as the models, in contrast with the point representativity of local observations."

R: We refer to climate data as input to simulation models, such hydrological or water resources models, for instance. We have made it clearer in the new version.

Q: "Line 50: Several more recent datasets have been ignored or not included in the text. In particular, at least SAFRAN-Spain (Quintana-Seguí et al. 2016, 2017), Iberia01 (Herrera

et al. 2019) and AEMET-Spain (Peral et al 2017), that have more recent periods and higher resolution than the ones cited by the authors."

R: We have added the missing references, thanks for the suggestion.

Q: "Lines 57-60: The reference for the observational datasets used have changed with respect to the abstract. Is there any reason?"

R: It is true. We have used the same exact wording in order to not create confusion on the reader.

Q: "Line 80: These two sentences seem to be contradictory. I mean, there is a manual quality control but the observations are automatically flagged as suspicious. There is a bold text and a reference to Table 1 but seems not being the caption of this table but normal text."

R: The flagging was made algorithmically, but the checks were done manually. We have re-phrased in a more precise way.

Q: "Evaluation: Although coherent, the notation is a litle confuse in my opinion. How do the authors define the indicator function? Usually, the 1 corresponds to the occurrence of precipitation and, then, the true positives are defined as obs > 0 and pred > 0. However, the authors use the opposite notation so I suppose that the indicator function identifies the days with 0 precipitation. This is in agreement with your equations, simply I commonly use another definition."

R: It all depends on which event we focus. Here we focus on the event pcp = 0. This is usual when evaluating classification models where the event being predicted is the less prevalent condition in the sample (e.g. being pregnant, or having a disease). We could have done it the other way (i.e., focus on predicting the chance of the event pcp > 0), but then the interpretation of all the binary evaluation statistics (precision, recall, sensitivity, specificity) will be reversed with respect to their usual meaning in the literature. In this trade-off, we opted to follow the conventions of classification model evaluation literature. We tried to be very precise and clear in defining the event being modelled, and we are certain that the current phrasing will allow the readers to understand the text.

Q: "Which is the range of the different parameters defined? I mean, PPV is lower than 1 so high values correspond to values close to 1. Similarly, the others parameters have their own range and interpretation."

R: All the statistic used to evaluate the binary model vary between 0 and 1, with values close to one being best. We have made it explicit in the new version, since it was not clear in the original version.

Q: "Results: Section 3.1,: The number of stations is referred to monthly data? The nature of the data is relevant to properly interpret the added value of the yuearbooks with

respect to the use only of the BNDC. It is surprising to me that, including in the modern years, the yearbooks could have this clear added value with respect the BNDC."

R: The yearbooks represented a very relevant improvement of the dataset, as we have tried to illustrate in Figure 1. The data shown is the number of stations available, and not observations (station-months). We have tried to make it clearer in the text.

Q: "The result about the mean distance to the closest station is reflecting that there is not any station isolated more than the dataset is spatiallu homogeneous. For example, suppose you have a dataset with 6 stations, three on the north and thre on the south of the Iberian Peninsula. If you remove one of each region, the mean distance doesn't change significantly, which is the result you obtain. I mean, is the reduction of the stations the one that is spatially homogeneous, avoiding any isolation of the stations. The effect of the stations reduction is more visible on the increment of the variability of that distance."

R: We agree in general with this remark. In fact, the variability is also showcased in the figure (Figure 4), and shows that there is a reduction in the distance variance with time, most notably between 1916 and 1950. We have incorporated this idea in the description of the results.

Q: "The correlation between precipitation and orography is high when monthly of climatological values are considered but it is not so evidente at shorter time scales."

R: We have added 'monthly precipitation' to make it more evident.

Q: "Section 3.2,: What is the meaning of the numbers inside the maps? Are they the points to which the authors refer after?"

R: The numbers identify the points that were selected for further analysis. It is explained in the figure caption, but we have added now a reference to the numbers to make it more clear.

Q: "The forumulation of the standard deviation of the kriging leads to the linear relation pointed out by the authors, Have they used others definitions for the uncertainty estimation? For example, the one proposed by Yamamoto (Yamamoto 2000)."

R: We have only considered the kriging's variance in this case. Other uncertainty estimators such as Yamamoto's are certainly interesting, but how to use them in the context of universal kriging is not evident. In any case, to test the interpolation quality (both in time and space) we have used a cross-validation approach and a number of statistics.

Q: "Section 3.3,: Could be the underestimation of the zero-precipitation frequency considered a common problem of gridded datasets?"

R: This is an interesting question, as accurately predicting zero-precipitation has proven challenging. We believe the problem is intrinsic to the nature of the problem, so yes, it is most likely a prevalent issue in most (if not all) gridded datasets. In many occasions, however, the accuracy of zero-precipitation is not tested, so it is difficult to reach any conclusion without a dedicated research.

Q: "Section 3.4: ".. variance contraction is expected in any interpolation ..." instead "is to be". Could be also "should be"."

R: Thanks, we have changed the sentence accordingly.

Q: "The parameter considered, MAE, suffers less than the RMSE of the inflation due to low number of anomalous values so the reasoning proposed would be more clear for that index instead the MAE."

R: While, on the other hand, the RMSE can be said to over-represent a low number of high residuals. Although we calculated an even higher number of error statistics, we opted to restrict the results to just one statistic for each relevant error type. We choose the MAE because it can be interpreted in the same units as the original variable. The results with the RMSE were fairly similar to those of the MAE.

Q: "The figure shown only include one year and does not explain the relative error of 41% obtained for July. How this error is in comparison with others existing datasets?"

R: Mentioning the relative error was a mistake, as it does not have much sense for a highly skewed variable such as precipitation, where the mean of the variable has little meaning. The relative error is very high in the summer months simply because the mean precipitation is very low in many stations, so even a low absolute error translates into a very high relative error. Consider, for example, a case where the mean is 1 mm, and the prediction is 2 mm. Considering the variability of precipitation this is a small error, but the relative error would be 200%. Therefore, we have opted for eliminating the reference to relative errors (also, relative errors are only mentioned on this single paragraph, and are not used anywhere else in the article).

Regarding to the comment about Figure 13 showing only one year, we provided it as an example, since it is not possible to show results for all the individual years in the data set. For that, we have provided Tables 5 to 7 and Figures 11 to 13 that contain validation statistics for the whole data set, as well as annual time series.

As for the high relative error in the summer months (highest in July), this result is highly conditioned by the misrepresentation of very low (zero, or near zero) values, which are more prevalent during the summer. This can be seen by the accumulation of dots along the axis in the figure. We have not compared the error magnitude with other existing datasets.

---

## Author Response (AR2)

**Author's response to the topical editor's comments.**

We submit a revised version of the manuscript that tackles all the issues raised by the topical editor.

In particular, we have unified the denomination of the official meteorological database of Spain using the English acronym (NCDB), instead of messing with two different acronyms in English and Spanish. We have also corrected the identified typo (GNDC was obviously a mistake for BNDC; it has been replaced by NCBD now).

We also added author contribution and competing interests sections, which were not present in the original submission.

We have identified several edition problems caused by the PDF conversion tool in MS Word. For instance, the duplicated and misplaced figure (Figure 8), and other minor problems such as broken lines and changes in text formatting. We have not found a good solution to this bug, but we have been able to solve the problem by replacing all internal references in the text (to figures and tables) by standard text. We suspect that the problems mentioned (but not specified) in lines 188, 203, and 446 are also related to the PDF conversion, but we were not able to find any issues on said lines.

Finally, regarding the criteria to select the figures and tables for the Appendix, we carefully selected those that were compulsory to illustrate the results described in the text, and we maintained them in the main body of the article. Figures and tables that contained additional information not necessarily required to follow the text were moved to the appendix, in order to not have too many in the main body. This included all the figures and tables related to the discussion of different options that were finally discarded (discussion section). We would prefer to maintain this criteria, since moving all the tables and figures with validation statistics to the appendix (as suggested) would leave the main body of the article with no content related to the dataset evaluation, which in our opinion is highly important.